# Multiplexed Optical Sensors in Arrayed Islands of Cells for multimodal recordings of cellular physiology

Christopher A. Werley [1,5], Stefano Boccardo[1,6], Alessandra Rigamonti[2], Emil M. Hansson[2,3] & Adam E. Cohen [1,4✉]

Cells typically respond to chemical or physical perturbations via complex signaling cascades which can simultaneously affect multiple physiological parameters, such as membrane voltage, calcium, pH, and redox potential. Protein-based fluorescent sensors can report many of these parameters, but spectral overlap prevents more than ~4 modalities from being recorded in parallel. Here we introduce the technique, MOSAIC, Multiplexed Optical Sensors in Arrayed Islands of Cells, where patterning of fluorescent sensor-encoding lentiviral vectors with a microarray printer enables parallel recording of multiple modalities. We demonstrate simultaneous recordings from 20 sensors in parallel in human embryonic kidney (HEK293) cells and in human induced pluripotent stem cell-derived cardiomyocytes (hiPSC-CMs), and we describe responses to metabolic and pharmacological perturbations. Together, these results show that MOSAIC can provide rich multi-modal data on complex physiological responses in multiple cell types.

[1] Department of Chemistry and Chemical Biology, Harvard University, Cambridge, MA 02138, USA. [2] Integrated Cardio Metabolic Centre, Department of Medicine Huddinge, Karolinska Institute, Huddinge, Sweden. [3] Department of Stem Cell and Regenerative Biology, Harvard University, Cambridge, MA 02138, USA. [4] Howard Hughes Medical Institute, Chevy Chase, MD 20815, USA. [5] Present address: Q-State Biosciences, Cambridge, MA 02139, USA. [6] Present address: Nobel Biocare AG, Kloten, Switzerland. ✉email: cohen@chemistry.harvard.edu

Starting with expression of green fluorescent protein (GFP) in model organisms in 1994[1], there has been continuous development of new genetically encoded fluorescent sensors[2–4]. A recent list catalogs 152 different sensors[5]. There are sensors for pH[6], calcium[7], redox potential[8], transmembrane voltage[9], second messenger signaling such as cyclic AMP[10] and diacyl glyceride (DAG)[11,12], and kinase activity such as ERK[13] or PKC[14], and protein activation such as Ras[15], RhoA[16], or Akt/PKB[17]. Because these sensors are genetically encoded, protein trafficking motifs can be added to target sensors to subcellular locations such as mitochondria[8,18,19], endoplasmic reticulum[20,21], or nucleus[22], leading to a combinatorial explosion of sensing modalities.

When a cell responds to external cues, multiple second messengers may be involved, and the cell state can change in many ways. To follow these complex, correlated dynamics, it is desirable to record from many fluorescent sensors simultaneously. The broad absorption and emission spectra of fluorescent proteins limit the number of independent color bands to ~4, which sets the maximum number of sensors that can be read in parallel from one cell. Multiplexing is further impeded by the fact that most fluorescent sensors use the GFP spectral channel.

To overcome this limitation, we developed MOSAIC: Multiplexed Optical Sensors in Arrayed Islands of Cells. MOSAIC uses densely packed arrays of islands of cells, each expressing a different fluorescent sensor (Fig. 1a). Similar cellular microarray techniques have been developed for high-throughput genetic screens[23]. To prepare the sample, a high-titer lentivirus encoding each sensor is deposited on an activated substrate using a microarray printer (Fig. 1b). Cells are plated uniformly over the virus array, and cells settling on each island are transduced with the corresponding fluorescent sensor. During cell plating, virus remains adhered to the substrate[23], minimizing sensor cross-contamination between islands. The cell patterning enables spatial multiplexing: many sensors with identical fluorescent spectra but distinct spatial coordinates are recorded simultaneously. A perfusion system rapidly delivers ligands or drugs, and the correlated responses of each sensor to a stimulus are recorded.

To make fluorescent recordings from the MOSAIC array, we used the Firefly microscope[24] (Supplementary Fig. 1), recently developed for wide-field imaging of electrical activity in neurons or cardiomyocytes. The microscope has a 6 × 6 mm field of view (FOV) that can be recorded with a frame rate of 100 Hz and a numerical aperture (NA) of 0.5 for high light collection efficiency. We have recorded from 6 × 7 island arrays with a 0.5 mm period. In principle, the Firefly FOV could record from a 12 × 12 array, tracking up to 144 different sensors simultaneously. Each spot of the array is 0.3 × 0.3 mm and contains tens to hundreds of cells, depending on cell size. The microscope has ~3 μm spatial resolution, so it is possible to identify individual cells to look for heterogeneity of response, or to average over many cells to determine the population behavior. Using a perfusion system (Supplementary Fig. 1), we can rapidly exchange media to assess responses to different types of perturbations.

There are several advantages to having all the reporters in the same well, as opposed to having each reporter in a different well of a multi-well plate. (1) Cells experience the same intervention, such as temperature change or ligand addition, at precisely the same time. It is possible to follow time-correlated responses of different sensors. (2) Cells have the same life history, avoiding well-to-well variability. When comparing the relative temporal responses of different sensors, e.g., when determining causality in a signaling cascade, it is critical that the starting cell states be as similar and controlled as possible. (3) Most sensors are also modulated by pH, which often changes in response to physiological perturbations. Concurrent pH measurements enable corrections of sensor responses for pH effects. (4) For cells that communicate over long distances via electrical signaling such as a cardiomyocytes or neurons, behavior can be synchronized throughout the well. Here, we demonstrate simultaneous recording of action potentials and calcium transients in the cytosol, mitochondria, and endoplasmic reticulum (ER) in a synchronized cardiomyocyte syncytium. (5) Cellular microarrays minimize the usage of cells and reagents, which can be critical for expensive primary or stem cell-derived samples.

First we describe the library of sensors (Table 1, Fig. 2). We then describe the patterning technology and implement it with human embryonic kidney 293T (HEK293) cells (Fig. 3). Next, we explain how to correct the nearly ubiquitous pH response of the fluorescent sensors (Fig. 4). We perform MOSAIC measurements of HEK293 cell responses to manipulations of cellular metabolism (Fig. 5). Finally, we apply the MOSAIC technique to human induced pluripotent stem cell-derived cardiomyocytes (hiPSC-CMs) (Figs. 6 and 7). We show high-speed recordings of voltage and calcium dynamics, as well as longer-timescale changes in cellular physiology in response to β-adrenergic agonists and blockers.

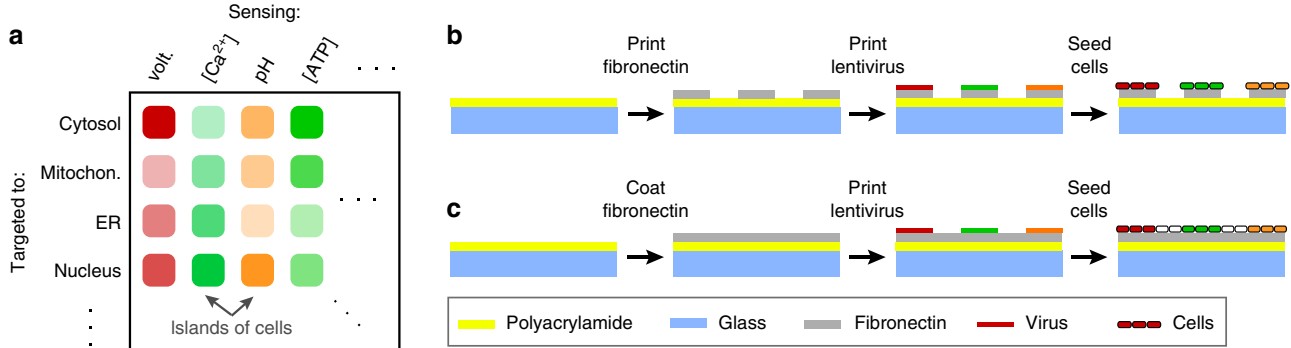

**Fig. 1 MOSAIC concept and patterning. a** Cells are patterned in arrayed islands, each expressing a genetically encoded fluorescent sensor targeted to a subcellular location. This approach enables simultaneous readout of many sensors. **b, c** Producing the MOSAIC arrays. **b** Highly motile cells, such as HEK293 cells, must be confined by patterning the substrate or they will move from their starting locations. Fibronectin, patterned into islands using a microarray printer, is covalently bound to a cell-repellent polyacrylamide surface. High-titer lentiviruses, each encoding a different fluorescent reporter, are spotted directly onto the fibronectin islands using the microarray printer. Cells are seeded throughout the dish, but they adhere exclusively to the fibronectin islands where they are infected by the virus and begin to express the fluorescent sensors. **c** For non-motile cells, such as cardiomyocytes, the fibronectin can be deposited homogeneously. This enables the development of an electrically connected syncytium that beats synchronously.

**Table 1 Library of MOSAIC fluorescent sensors.**

| Sensor name | Detected quantity | Subcellular localization | Plasmid ID number | Calculated quantity $S$ | Ex. $\lambda$ (nm) | Em. $\lambda$ (nm) | Supp. Fig. | Refs. |
|---|---|---|---|---|---|---|---|---|
| QuasAr2 | Voltage | Membrane | pMOS009 | $F/F_0$ | 635 | 710 | 2 | 9,70 |
| CheRiff | Stimulation | Membrane | pMOS010 | — | 470 | — | 3 | 9,71 |
| GCaMP6F | $Ca^{2+}$ | Cytosol | pMOS008 | $F/F_0$ | 488 | 512 | 4 | 7,39 |
| Mitycam | $Ca^{2+}$ | Mito. | pMOS028 | $2 - F/F_0$ | 513 | 530 | 5 | 18,72–74 |
| GCaMPer | $Ca^{2+}$ | ER | pMOS003 | $F/F_0$ | 488 | 512 | 6 | 21 |
| ClopHensorN | $Cl^-$ | Cytosol | pMOS019 | $F/F_0$ | 488 | 512 | 7 | 75,76 |
| Superecliptic pHluorin | pH | Cytosol | pMOS005 | $F/F_0$ | 488 | 512 | 8 | 6,69 |
| Superecliptic pHluorin | pH | Mito. | pMOS011 | $F/F_0$ | 488 | 512 | 9 | 6,69 |
| Superecliptic pHluorin | pH | ER | pMOS031 | $F/F_0$ | 488 | 512 | 10 | 6,69 |
| Ratiometric pHluorin | pH | Cytosol | pMOS017 | $\frac{F^{405}F_0^{488}}{F_0^{405}F^{488}}$ | 405, 488 | 512 | 11 | 6,69 |
| Ratiometric pHluorin | pH | Mito. | pMOS018 | $\frac{F^{405}F_0^{488}}{F_0^{405}F^{488}}$ | 405, 488 | 512 | 12 | 6,69 |
| Ratiometric pHluorin | pH | ER | pMOS030 | $\frac{F^{405}F_0^{488}}{F_0^{405}F^{488}}$ | 405, 488 | 512 | 13 | 6,69 |
| Grx1-roGFP2 | Glutathione redox | Cytosol | pMOS014 | $\frac{F^{405}F_0^{488}}{F_0^{405}F^{488}}$ | 405, 488 | 512 | 14 | 8,77 |
| Mito-roGFP2-Grx1 | Glutathione redox | Mito. | pMOS013 | $\frac{F^{405}F_0^{488}}{F_0^{405}F^{488}}$ | 405, 488 | 512 | 15 | 8,77,78 |
| Grx1-roGFP1-iE$_{ER}$ | Glutathione redox | ER | pMOS012 | $\frac{F^{405}F_0^{488}}{F_0^{405}F^{488}}$ | 405, 488 | 512 | 16 | 20 |
| roGFP2-Orp1 | $H_2O_2$ oxidation | Cytosol | pMOS016 | $\frac{F^{405}F_0^{488}}{F_0^{405}F^{488}}$ | 405, 488 | 512 | 17 | 19,77 |
| Mito-roGFP2-Orp1 | $H_2O_2$ oxidation | Mito. | pMOS015 | $\frac{F^{405}F_0^{488}}{F_0^{405}F^{488}}$ | 405, 488 | 512 | 18 | 19,77 |
| Peredox | NADH/NAD$^+$ | Cytosol | pMOS023 | $F/F_0$ | 405 | 512 | 19 | 28,79 |
| PercevalHR | ATP/ADP | Cytosol | pMOS007 | $\frac{F^{405}F_0^{488}}{F_0^{405}F^{488}}$ | 405, 488 | 512 | 20 | 80 |
| FLII$^{12}$Pglu-700$\mu\delta6$ | Glucose | Cytosol | pMOS020 | $\frac{F^{405}F_0^{488}}{F_0^{405}F^{488}}$ | 440 | 480, 530 | 21 | 81–83 |
| Upward DAG | Diacyl glyceride | Cytosol | pMOS024 | $F/F_0$ | 488 | 512 | 22 | 11,12 |

Plasmid identification (ID) number is used to reference sensors throughout the manuscript.

## Results

**Assembly of MOSAIC sensor library**. A critical first step for MOSAIC was assembling the collection of fluorescent protein-based sensors (Table 1 and Supplementary Table 1), generously shared by many labs (see Acknowledgements). For this initial demonstration, we used 20 sensors detecting key physiological parameters. For each sensor, we cloned the coding DNA into a third-generation lentiviral backbone under control of the cytomegalovirus (CMV) universal promoter (Methods). We prepared high-titer lentivirus via ultra-centrifugation and reserved several aliquots of unconcentrated lentivirus for sensor validation and testing.

Each fluorescent sensor was validated in iPSC-derived cardiomyocytes and many were also validated in HEK293 cells. These results, and a brief overview of each sensor, are shown in Supplementary Figs. 2–22. Summary data on sensor brightness, sensitivity, and pH response are in Supplementary Table 1. A perfusion system was used for rapid compound delivery and buffer exchange, with a protocol designed for each sensor to highlight the detected physiological response. For sensors targeted to subcellular locations, we confirmed trafficking with a live-cell stain for mitochondria or endoplasmic reticulum (Methods).

Data for four example sensors in hiPSC-CM's are shown in Fig. 2. Figure 2a shows mitochondrial trafficking of the sensor mito-roGFP2-Grx1[8], which detects the glutathione redox potential. Figure 2b shows the response to rapid intracellular oxidation with hydrogen peroxide followed by rapid reduction with dithiothreitol (DTT). Mito-roGFP2-Grx1 is a ratiometric sensor with one fluorescent emission channel peaked at ~510 nm and two excitations at 405 nm (purple trace) and 485 nm (teal trace). The sensor output $S$ (black trace), here the ratio of the purple and teal traces but defined for each sensor in Table 1, reports overall readout. $\Delta S/S_0 = (S - S_0)/S_0$, with $S_0$ the baseline signal level in pH 7.4 imaging buffer, is a measure of effect size. The nearly twofold change in $\Delta S/S_0$ in response to $H_2O_2$, and the minimal drop ($-8\%$ $\Delta S/S_0$) below baseline after addition of DTT, demonstrates the highly reduced initial state of the mitochondrial matrix.

Figure 2c shows trafficking of the ER-targeted glutathione redox sensor Grx1-roGFP1-iE$_{ER}$[20], optimized to function in the more oxidized environment of the ER. The sensor showed a more modest response to $H_2O_2$ (23% $\Delta S/S_0$) but dropped far below baseline ($-27\%$ $\Delta S/S_0$) when the ER was fully reduced with DTT. Furthermore, recovery to baseline levels after washing with imaging buffer was slow ($\gg1$ min).

Figure 2e shows calcium transients detected with GCaMP6F[7] as the electrically connected cardiomyocyte syncytium beat spontaneously. Figure 2f shows the upward DAG fluorescent sensor[11], which reports on the second messenger diacylglycerol from the PIP$_2$/PLC/DAG signaling cascade. A strong response (43% $\Delta F/F_0$) resulted from triggering the signaling cascade with carbachol, an acetylcholine analog that binds muscarinic and nicotinic acetylcholine receptors (mAChRs and nAChRs)[25]. The M3 mAChR is a G$_q$-coupled G protein-coupled receptor (GPCR) expressed in cardiomyocytes that triggers the PIP$_2$/PLC/DAG signaling cascade[26]. This GPCR is implicated in regulation of potassium channel current and action potential shape, connexins and gap junctions, anti-apoptotic signaling, and calcium handling[27].

**Printing the MOSAIC array**. We prepared MOSAIC arrays with a microarray printer for experiments with HEK293 cells and

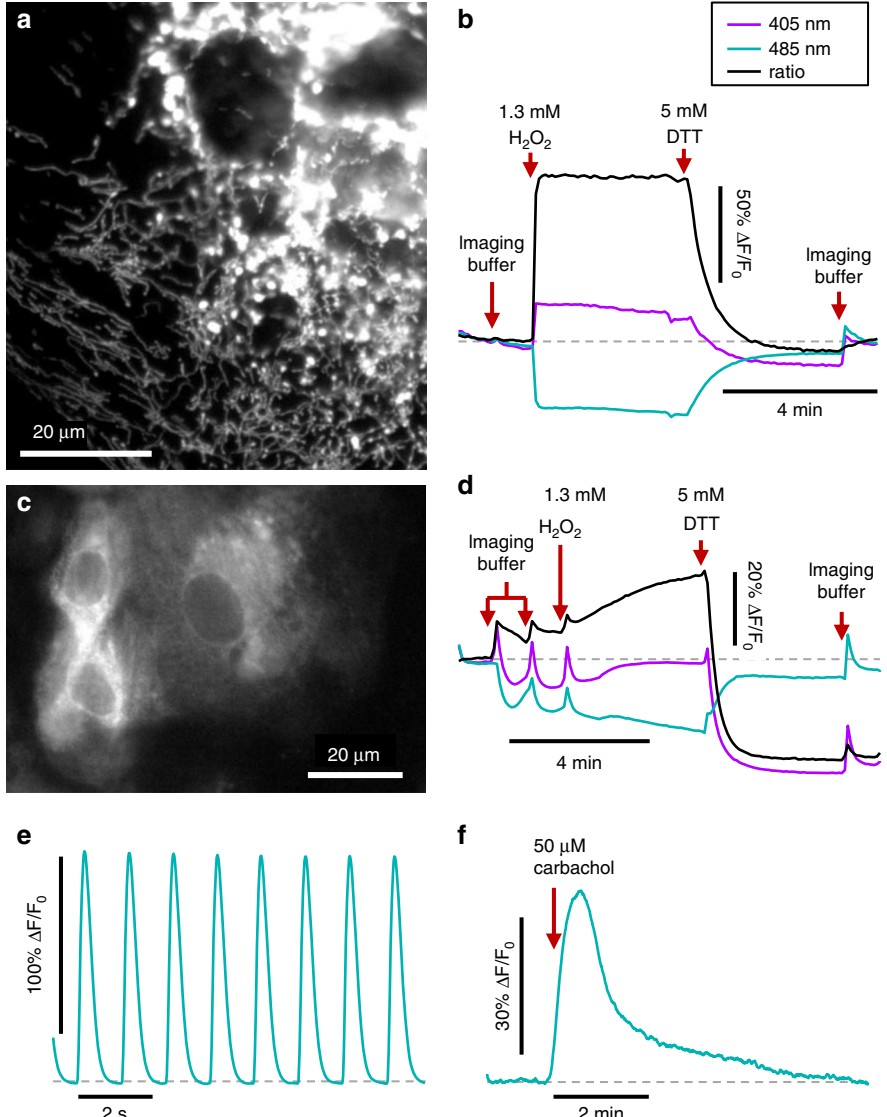

**Fig. 2 Example MOSAIC sensors in hiPSC-derived cardiomyocytes. a** Fluorescence image of mitochondrial redox sensor mito-roGFP2-Grx1 in hiPSC-CMs. Image γ was set to 0.75 to highlight dim mitochondria in cell periphery. **b** Fluorescence responses of mito-roGFP2-Grx1 to changes in redox state induced by hydrogen peroxide or DTT. The sensor readout (black) is calculated by the ratio of the fluorescence with 405 nm illumination (purple) to 485 nm illumination (teal). **c** Fluorescence image of ER redox sensor Grx1-roGFP1-iE$_{ER}$ in hiPSC-CMs. **d** Fluorescence response of Grx1-roGFP1-iE$_{ER}$ to changes in redox state induced by hydrogen peroxide and dithiothreitol (DTT). **e** Fluorescence dynamics of cytosolic calcium sensor GCaMP6F during spontaneous beating of the cardiomyocyte syncytium. **f** Fluorescence response of diacylglycerol sensor upward-DAG to cholinergic stimulation with carbachol.

cardiomyocytes. For HEK293 cells, which are motile and can move away from their initial site of viral transduction, it was necessary to confine the cells to islands. We prepared glass-bottomed dishes with a polyacrylamide gel, which prevents cell adhesion (Methods). The polyacrylamide was functionalized with N-hydroxysuccinimide esters, which allowed rapid and irreversible covalent bonding with primary amines. A microarray printer was used to deposit an array of $300 \times 300\,\mu m$ islands of fibronectin with an array period of 500 μm. Primary lysines in the fibronectin covalently bound to the NHS-functionalized polyacrylamide gel, making a stable substrate for cell growth. The white-light image in Fig. 3f and magnified view of the cells in (Fig. 3o) demonstrates the effective cell patterning with high viability on the fibronectin and no migration onto the bare polyacrylamide. With extended culture of the HEK293 cells, they grew into multi-layers (e.g., Fig. 3f, column 1, rows 4–6) but remained confined to the patterned squares.

Using the same microarray printer, we deposited different high-titer lentiviruses directly on top of the fibronectin islands. The viral printing buffer, developed in ref. [23], contained trehalose and glycerol to maximize lentivirus survival as the nanoliter-volume printed spot dried. Once MOSAIC arrays were printed, they were immediately stored on ice. Cells were plated in the glass bottom dish the same day; freezing of MOSAIC arrays for long-term storage resulted in substantial reduction in transduction efficiency.

Figure 3 shows HEK293 cells deposited on a MOSAIC array. Each fluorescent sensor was printed on a pair of adjacent islands to facilitate quantification of island-to-island variability. To accommodate the distinct spectra of the different reporters, we imaged the entire array with three pairs of excitation and emission wavelengths (Fig. 3a–e). Images were captured with the Firefly microscope[24] shown in Supplementary Fig. 1, which can collect fluorescence from a large field of view with high efficiency.

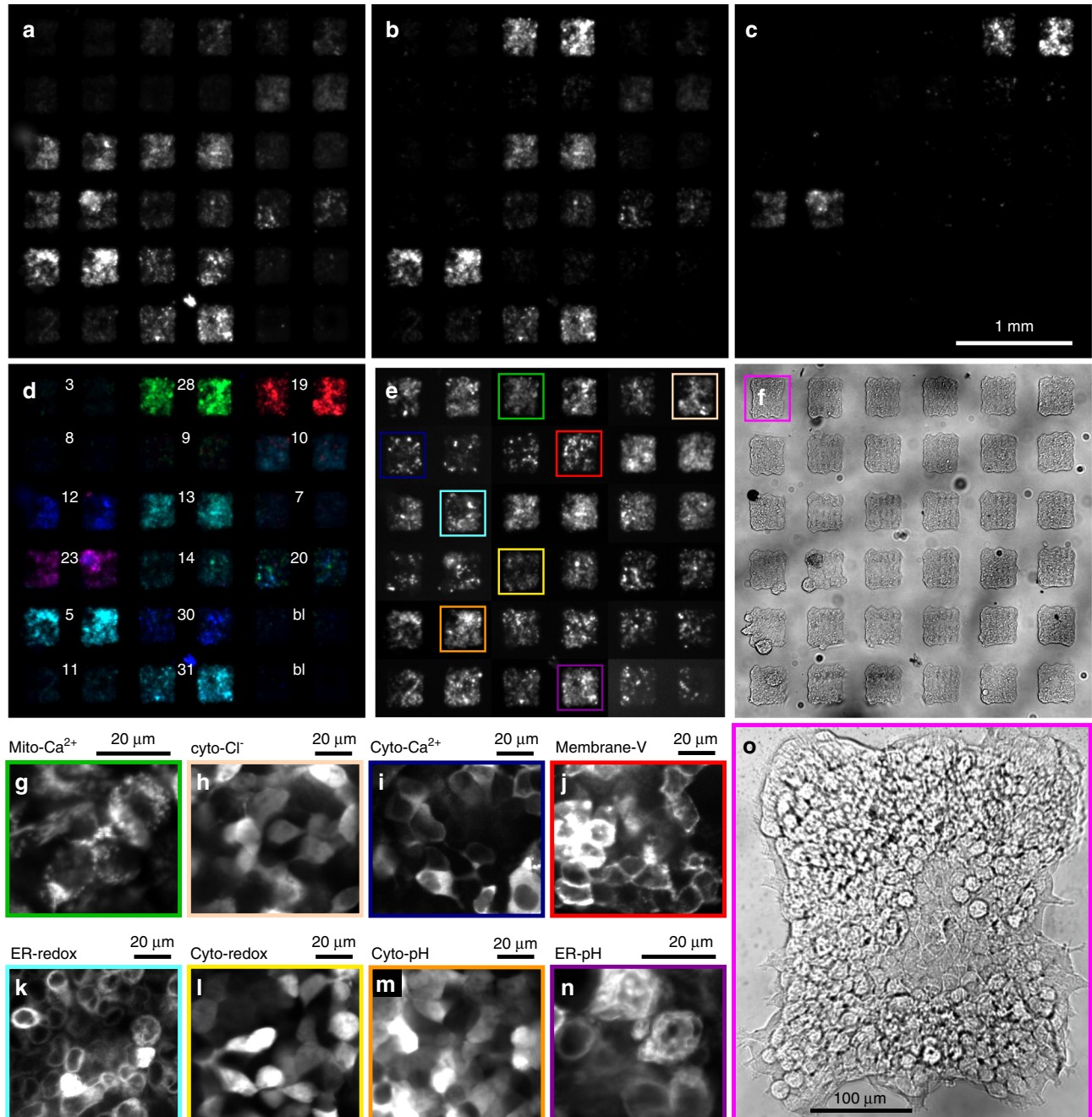

**Fig. 3 HEK293 cell MOSAIC arrays.** Fibronectin was patterned with a microarray printer to define islands. A lentivirus coding for different fluorescent sensors was printed on top. HEK293 cells were plated across the dish; they attached to each island and were infected by the corresponding virus. The array is 6 × 6 islands with adjacent islands containing identical sensors. **a** wtGFP fluorescence channel (405 nm Exc, 540/50 nm Em). **b** EGFP fluorescence channel (488 nm Exc, 540/50 nm Em). **c** tdTomato/mCherry fluorescence channel (565 nm Exc, 625/30 nm Em). **d** Merge of the fluorescence channels: **a** blue, **b** green, and **c** red. The numbers indicate the plasmid ID number, see Table 1. bl = blank, no virus spotted. Blue islands are based on sensors spectrally similar to wild-type GFP, cyan islands are based on EGFP, green islands are based on EYFP, magenta islands contain a redox sensor based on a sapphire-mCherry fusion, and red islands have a Cl-sensor based on an EGFP-tdTomato fusion. **e** The same image as **b** but with contrast adjusted for each island pair to account for brightness differences between sensors. **f** White-light trans-illumination image. **g**–**n** High-magnification spinning disk confocal images from sub-regions of the islands boxed in **e**. **g** Mitochondrially targeted calcium sensor Mitycam. **h** Cytosolically-targeted chloride sensor ClopHensorN. **i** Cytosolically-targeted calcium sensor GCaMP6F. **j** Cell membrane-targeted voltage sensor QuasAr2-mOrange2. **k** ER-targeted redox sensor Grx1-roGFP1-iE$_{ER}$. **l** Cytosolically-targeted redox sensor Grx1-roGFP2. **m** Cytosolically-targeted pH sensor supereclptic pHluorin. **n** Endoplasmic reticulum-targeted pH sensor ER-SE pHluorin. **o** Magnified transmitted-light view of island boxed in **f**.

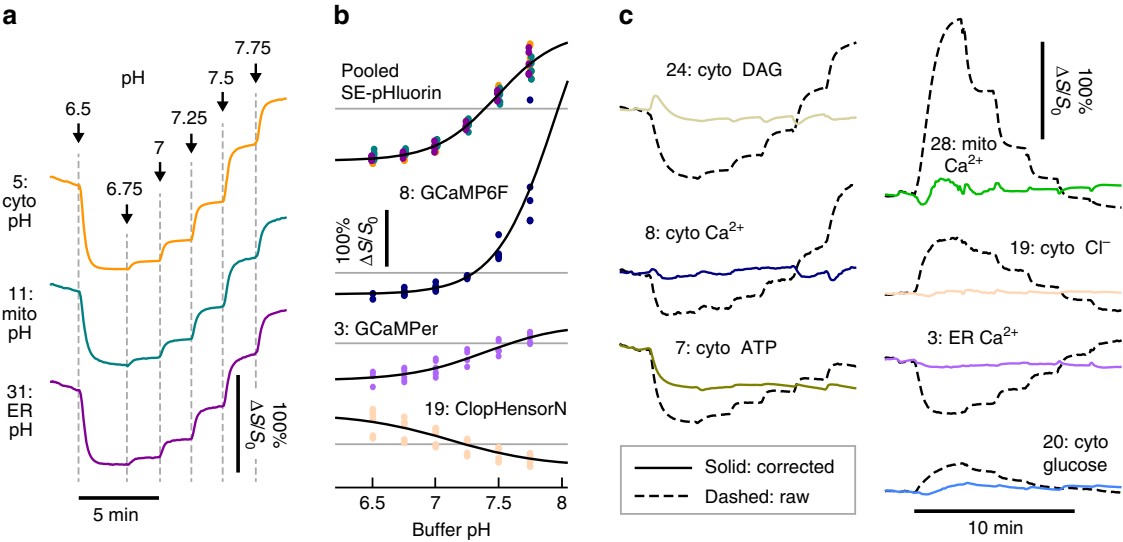

**Fig. 4 Correction of pH-induced fluorescence changes in HEK293 cells.** In a full MOSAIC array, the pH was changed sequentially from 6.5 to 7.75 while recording from all the sensors. To equilibrate the intracellular pH with the perfusate, the buffer contained the $K^+/H^+$ exchanger nigericin (14 μM) and high potassium (100 mM $K^+$) to dissipate proton and potassium gradients. Plasmid ID numbers precede the label for each trace. **a** Fluorescence recordings from supereclliptic pHluorin sensors in three subcellular locations. The signal level $S$ is defined for each sensor in Table 1. **b** Fluorescence as a function of pH for four example sensors (points) and fits to a Hill equation (black line). The fit values are tabulated in Supplementary Table 1. Gray lines represent baseline signal in pH 7.4 buffer. The dots represent $n = 3$ independent experimental rounds with array printing and cell plating on different days. The SE-pHluorin fit was calculated by pooling subcellular location; dots are color coded to traces in **a**. **c** Raw recordings from each fluorescent sensor, calculated according to the equation in Table 1, before (dashed black) and after (solid color) pH correction.

Sensors localized to the mitochondria, ER, cytosol, and cell membrane showed distinct patterns of expression.

**Functional recordings in HEK293 cells and pH corrections**. Functional recordings were made with alternating excitation at 405 and 485 nm, and emission collected in the GFP channel (Em: 515–565 nm). Ratiometric signals (e.g., Fig. 2b, d) were calculated from both images while non-ratiometric sensors (e.g., Fig. 2e, f) used only the 485 nm excitation images. Paired images were typically recorded every 5 or 10 s. Timing and synchronization between the camera, illumination light, and perfusion system were managed with a data acquisition (DAQ) card and custom LabView control, resulting in microsecond timing jitter (Methods).

Whole-island fluorescence signals were calculated from a weighted average of the corresponding pixels, with pixel weights selected to maximize SNR and to reject occasional cells that lifted off one island and deposited on a neighboring island (Methods). The signal trace, $S$, for each sensor was calculated according to Table 1. Example traces from HEK293 cells are shown in Figs. 4 and 5, and example traces from cardiomyocytes are shown in Fig. 7.

The fluorescence of many protein-based sensors is pH dependent. Typically, the pH sensitivity comes from the intrinsic pH sensitivity of the fluorescent protein, although the sensing domain may also respond to pH changes. These pH effects and our correction methodology are displayed in Fig. 4. Figure 4a shows the response of the supereclliptic pHluorin sensor[6] targeted to the cytosol, mitochondria, and ER in response to the perfusion of buffers with different pH. The cell membrane was permeabilized with the $K^+/H^+$ exchanger nigericin at 14 μM and the buffer was prepared with 100 mM $K^+$ to minimize the potassium gradient and to facilitate free proton exchange. Similar recordings were made in triplicate for each MOSAIC sensor. Examples are shown in dashed black in Fig. 4c. After the fluorescence level

stabilized following perfusion, the fluorescence at each pH was measured and tabulated (points in Fig. 4b), and the resulting titration was fit with the Hill equation (solid line in Fig. 4b). Fitting results are shown in Supplementary Table 1. The roGFP-based redox sensors and the $NADH/NAD^+$ sensor Peredox were not pH corrected. Peredox can be used without pH correction because it was engineered to minimize pH effects[28]. The ratiometric readout of the roGFP sensors also was minimally perturbed by pH changes: in the change from pH 7 to 7.25, where the redox state appeared stable, $\Delta S/S_0$ was 1%, 2%, and 0.5% for cyto-, mito-, and ER-Grx1-roGFP respectively, which is small compared with redox-induced fluorescent changes in these sensors of 50–200% (Fig. 5).

To correct for pH in MOSAIC recordings, the time-dependent pH in each cellular subcompartment was calculated by applying the inverted Hill fit to the subcompartment-targeted supereclliptic pHluorin recordings. The changes in fluorescence for each sensor attributable to pH were then calculated by applying the Hill fit for the corresponding sensor with scaling coefficients (Methods). Fluorescence variations not attributed to pH were ascribed to the intended sensing modality. Figure 4c shows example traces during the pH titration shown in (a) before (dashed black) and after (solid color) pH correction. Across the 7 pH-corrected sensors, pH changes between 6.5 and 7.75 induced r.m.s. changes in signal of 110% before correction, but only 4% after correction, demonstrating a substantial reduction in pH artifacts.

**Probing metabolic perturbations**. Figure 5 shows MOSAIC responses to a series of metabolic and redox perturbations, shown along the top of the figure, during a ~1-h recording. Washes are shown in red. For both HEK293 cells and cardiomyocytes, we began each protocol with two washes with unmodified imaging buffer to bring cells to controlled initial conditions. The responses during these washes were not analyzed. First, a mild stress was introduced by depriving the cells of the energy source (glucose

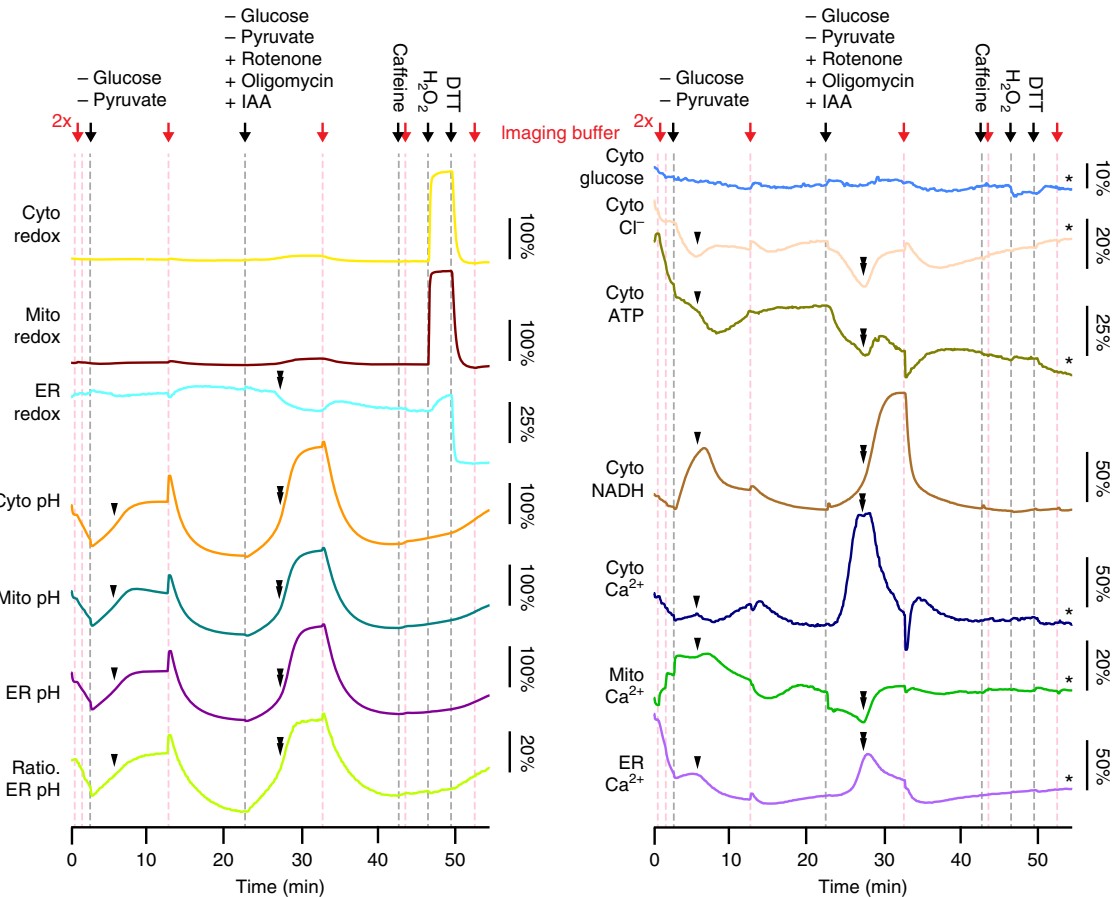

**Fig. 5 Full MOSAIC array recording in HEK293 cells.** MOSAIC reported responses to perturbations to cellular metabolism and redox state. Sensor identity is indicated on the left of each trace, and sensor response scale bar ($\Delta S/S_O$, see Table 1 for definition) on the right. Stars at the right side of the trace indicate that pH correction was performed. To manipulate cell metabolism, we first exchange the standard imaging buffer (10 mM galactose and 1 mM pyruvate) to a buffer free of energy sources (0 mM sugar, 0 mM pyruvate). After rinse and recovery, we washed with a buffer without energy sources and with compounds to arrest energy production: 2.5 μM oligomycin to block ATP synthase, 1 μM rotenone to block mitochondrial electron transport chain complex I, and 1 mM iodoacetic acid (IAA) to arrest glycolysis by interfering with glyceraldehyde 3-phosphate dehydrogenase. Disruption of metabolism caused correlated changes in pH, chloride, ATP, NADH, and calcium. We later washed with 5 mM caffeine, followed by 1.3 mM hydrogen peroxide to oxidize the cells, followed by 5 mM dithiothreitol (DTT) to reduce the cells. Caffeine did not have a strong effect, while the redox manipulations showed strong signatures in the redox sensors.

and pyruvate) for 10 min. As soon as the energy sources were removed, the NADH/NAD$^+$ sensor Peredox reported a rapid increase in the reduced form, NADH. This is likely explained by the drop in pyruvate and the lactate dehydrogenase (LDH)-catalyzed redox reaction between pyruvate and NADH: pyruvate + NADH + H$^+$ ⇌ NAD$^+$ + lactate, and is consistent with previous results[28]. Concurrent with the NADH rise, the pH in all cellular compartments increased and the ATP and chloride levels decreased.

Approximately 3 min into the first energy deprivation (Fig. 5, black arrowheads), there was a concerted change in many intracellular parameters, including mitochondrial and ER calcium, chloride, pH, NADH/NAD$^+$, and ATP. After the change, NADH/NAD$^+$ and ATP levels moved back toward baseline. These spontaneous changes suggest a metabolic transition toward increased oxidative phosphorylation, as follows. HEK293 cells cultured with excess glucose, as these were, normally have inefficient ATP generation, with only 22% of the pyruvate derived from glycolysis entering the citric acid cycle[29]. We hypothesize that the energy deprivation may have activated metabolism of cellular peptides and lipids to produce acetyl-CoA and other intermediates that can drive the citric acid cycle. This transition would directly replenish NADH and could restore ATP levels

through the mitochondrial electron transport chain and ATP synthase[30]. The observed increase in mitochondrial Ca$^{2+}$ is consistent with activation of aerobic respiration: increased mitochondrial Ca$^{2+}$ would upregulate both the citric acid cycle and oxidative phosphorylation[31]. After a subsequent wash with imaging buffer, most sensors returned to a level at or near their pre-starvation values.

Next, we again removed sugar and pyruvate, while adding a cocktail of drugs (rotenone, a mix of oligomycins A, B, and C, and iodoacetic acid) to arrest both glycolysis in the cytosol and aerobic respiration in the mitochondria. This challenge evoked large changes in many cellular parameters with dynamics on the minute timescale. There was a dramatic increase in cytosolic calcium, with correlated changes in pH, chloride, mitochondrial calcium, ER calcium, and ER redox (Fig. 5, black double-arrowheads). These results suggested that autophagy may have been triggered: autophagy is induced by nutrient deprivation to mobilize peptide and lipid cellular energy stores[32,33]. An autophagic flux assay showed increased accumulation of autophagosome marker LC3-II in this energy deprivation condition relative to vehicle control (Supplementary Fig. 23), confirming this hypothesis. Both an increase in cytosolic Ca$^{2+}$ and a decrease in mitochondrial Ca$^{2+}$, are associated with

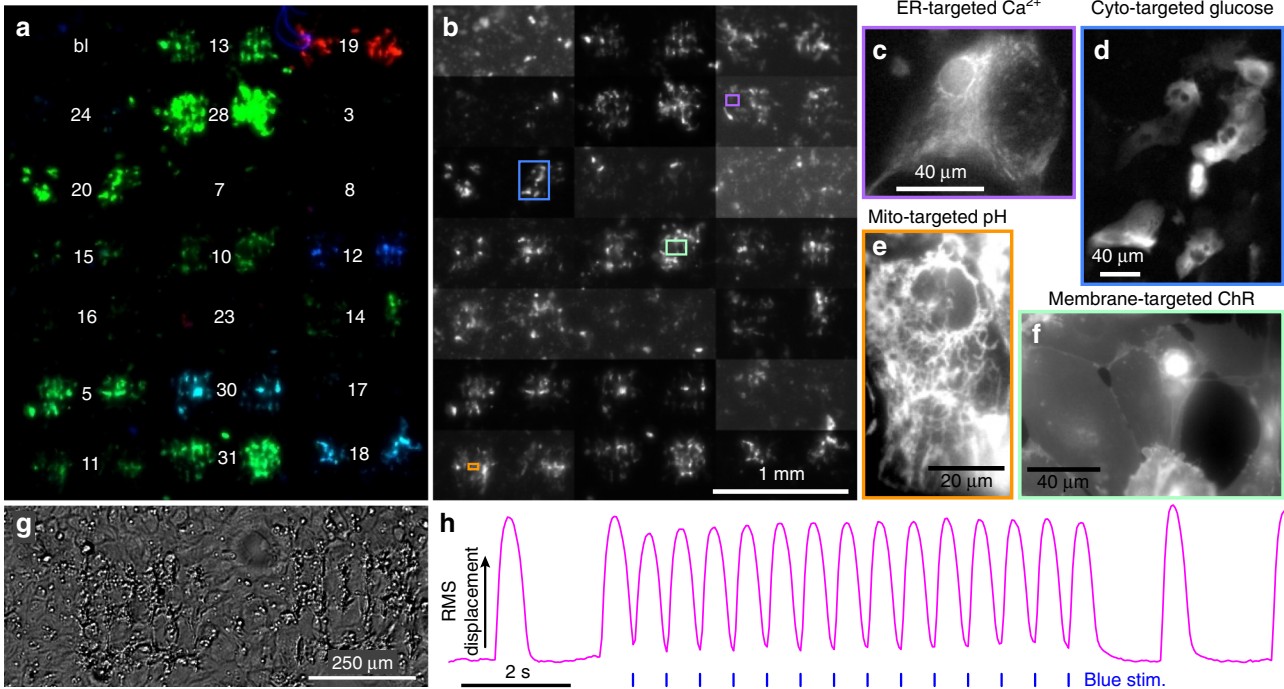

**Fig. 6 Cardiomyocyte MOSAIC arrays.** A cardiomyocyte syncytium was seeded on a microarray of lentivirus coding for various fluorescent sensors. In the 7 × 6 array, adjacent spots contained identical sensors. **a** A merged image of the different color channels. Blue: wtGFP fluorescence channel (405 nm Exc, 540/50 nm Em); green: EGFP fluorescence channel (488 nm Exc, 540/50 nm Em); red: tdTomato fluorescence channel (565 nm Exc, 625/30 nm Em). Island pairs are labeled with plasmid ID numbers, see Table 1. bl = blank, no virus spotted. Blue and cyan spots are based on sensors spectrally similar to wild-type GFP, green spots are sensors based on enhanced GFP or YFP, and red spots are fused with tdTomato or mCherry. **b** Green channel, with contrast adjusted for each island pair. Magnified views of the colored boxes are shown in **c–f**. **c** Endoplasmic reticulum-targeted calcium sensor GCaMPer. **d** Cytosolically-targeted glucose sensor FLII12Pglu-700uDelta6. **e** Mitochondrially targeted pH sensor mito-supercleptic pHluorin. **f** Cell membrane-targeted channelrhodopsin CheRiff fused with EGFP. **g** White-light image of the first two blank spots in the array upper left. Increased roughness in the printed region results from the microarray pin piercing the polyacrylamide gel substrate. **h** Beating of the cardiomyocyte syncytium revealed by motion in white-light images during a 1.8 Hz stimulus train delivered by blue light driving the CheRiff spots. The paced beats are preceded and followed by spontaneous beating.

induction of autophagy[34–37], consistent with observations from the MOSAIC array. We speculate that the transient calcium signals observed here may have triggered that signaling cascade. In contrast to the initial nutrient deprivation, here ATP and NADH could not recover to baseline levels, likely from the pharmacological block of aerobic respiration. The large increase in the NADH/NAD$^+$ ratio is consistent with block by rotenone of NADH oxidation by the complex I of the mitochondrial electron transport chain. With both glycolysis and aerobic respiration blocked, the cells had limited options for restoring ATP levels. As in the prior experiments, most parameters recovered to baseline values during the 10 min after a thorough wash in imaging buffer, although the ATP remained depressed.

A final set of interventions included 5 mM caffeine, followed by redox perturbations to exercise other sensors in the MOSAIC array. The caffeine showed minimal effects. Oxidation of the reduced environment in the cytosol and mitochondria with 1.3 mM $H_2O_2$ induced threefold increases in redox sensor $\Delta S/S_0$, but only induced a 10% $\Delta S/S_0$ change in the sensor reporting on the already oxidized environment in the ER. Subsequent reduction with 5 mM DTT returned the cytosolic and mitochondrial redox state to baseline while reducing the ER well beyond basal conditions and inducing a −32% $\Delta S/S_0$ drop below baseline. These large perturbations to cellular redox state evoked minimal changes in cellular pH, calcium, or metabolic parameters.

Supplementary Movie 1 is paired with Fig. 5 and shows the response for nine selected sensors. Within each island the cells showed largely homogeneous and concurrent responses, indicating minimal cross-contamination between islands and justifying sensor time trace calculation from whole-island averages.

**MOSAIC in human iPSC-derived cardiomyocytes.** Next we applied MOSAIC to hiPSC-CMs (Cellular Dynamics International, iCell cardiomyocytes), which are ~95% pure[38] and form functionally active syncytia with myofibril banding when cultured on polyacrylamide gel[39]. To enable formation of an extended, electrically coupled syncytium of cardiomyocytes that beat synchronously, we did not pattern the cells into discrete islands. Instead, fibronectin was deposited across the entire polyacrylamide gel, followed by patterning of the virus with the microarray printer (Fig. 1c). Supplementary Movie 2 shows a 48-h time-lapse video of cultured cardiomyocytes, where three spots with a 1 mm spacing were printed with the cyto-pH sensor (pMOS005). The transduction efficiency within the spots was high, and ~20 cardiomyocytes expressed the sensor in each spot. Except for a single highly motile fibroblast, most cells remained on their targeted islands.

Figure 6 shows MOSAIC arrays with cardiomyocytes. The cells expressed the patterned lentiviral constructs and remained localized throughout their 10–14 day culture (Fig. 6a, b). The transduction efficiency for many sensors was comparable to the cyto-pH discussed in the previous paragraph, but some lower-efficiency viruses such as the glucose sensor (pMOS020)

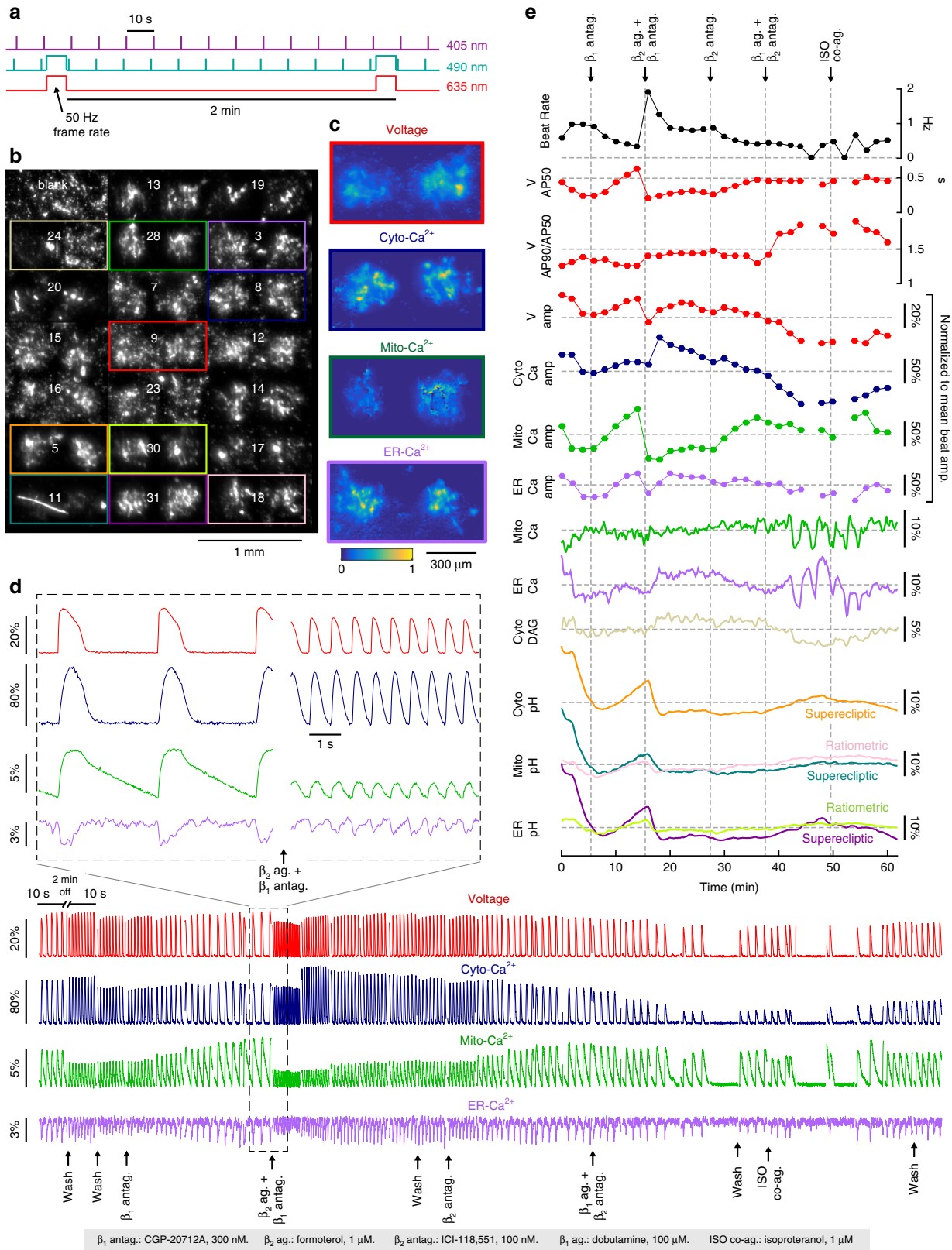

β1 antag.: CGP-20712A, 300 nM.  β2 ag.: formoterol, 1 µM.  β2 antag.: ICI-118,551, 100 nM.  β1 ag.: dobutamine, 100 µM.  ISO co-ag.: isoproteranol, 1 µM.

transduced <10 cardiomyocytes/island. Fluorescent sensors localized to appropriate subcellular compartments in the transduced cardiomyocytes (Fig. 6c–f). The entire field of view maintained synchronous beating (Fig. 6h). Motion in the white-light trans-illumination recording was reported by the root-mean-square of the difference between each frame and the average image acquired during a period without any beats. The two center islands contained the channelrhodopsin CheRiff[9], which allowed optical pacing of the entire syncytium (Fig. 6h).

Most MOSAIC sensors required blue excitation light for fluorescence imaging, which would have led to spurious activation of the CheRiff. To address this form of crosstalk, we

**Fig. 7 β-adrenergic stimulation of cardiomyocytes reveals subcompartment-specific physiological responses. a** Recordings interleaved time-lapse imaging with intervals of high-speed imaging to enable collection of signals at different time resolutions (Methods). **b** Average fluorescence image in the GFP channel with contrast adjusted for each sensor island. Sensor plasmid ID numbers are overlaid in white (see Table 1). Islands used in other figure panels are boxed with the corresponding colors. A small fiber contaminated the islands on the lower left. **c** Images of the normalized correlation between each pixel time-trace and the average 50 Hz recording time trace show active cardiomyocytes expressing the correct reporter. **d** Voltage, cytosolic calcium, mitochondrial, and ER calcium responses to beta-adrenergic stimulation during a 1 h recording. Acute drug responses were monitored immediately following a 10 s perfusion and during the following 10-min period before the next intervention. Baseline changes between recording epochs have been removed. There was little change in voltage and cytosolic calcium baseline; the changes in mitochondrial and ER calcium baselines are shown in **e**. Inset: a magnified view of the waveforms before and after addition of $\beta_2$ agonist in a background of $\beta_1$ antagonist, revealing marked changes in beat rate and beat shapes and amplitudes. **e** Physiological dynamics in different cellular compartments. The top seven traces (circles), sampled at 2-min intervals, are calculated from the beating waveforms in **d**. The voltage and calcium amplitude traces have been normalized to the mean beat amplitude across the recording. The bottom eight traces (solid lines) were calculated from image pairs (405 and 488 nm excitation) captured every 10 s throughout the 60 min recording. Calcium, second messenger diacyl glyceride (DAG), and superercliptic pHluorin recordings used the 488 nm excitation images exclusively. The ratiometric pHluorin images used the fluorescence ratio. The mitochondrial $Ca^{2+}$ and ER-$Ca^{2+}$ recordings have been corrected for pH changes via the corresponding subcompartment ratiometric pHluorin pH readout. DAG was corrected for pH changes using the SE-pHluorin. pH changes are clearly visible in the raw calcium and DAG time traces.

subsequently omitted CheRiff from the primary MOSAIC array. Instead, we printed a second array with all CheRiff islands separated by several millimeters from the primary array. A second blue light source mounted above the sample paced the CheRiff-expressing cells, which then synchronized the whole syncytium through gap junction-mediated conduction. The primary light source applied steady illumination to the MOSAIC array. Supplementary Fig. 24 shows images of two other MOSAIC arrays, demonstrating reproducible and high-quality cell patterning, gene expression, and voltage and calcium dynamics.

MOSAIC enables tracking multiple physiological parameters in cardiomyocytes with sub-beat temporal resolution. See for example the voltage, cytosolic $Ca^{2+}$, mitochondrial $Ca^{2+}$, and ER $Ca^{2+}$ recordings in Fig. 7d. In cultured cardiomyocyte syncytia, action potentials typically propagate at ~100 μm/ms[39,40], so a wave takes ~40 ms to traverse the 4 mm maximal diagonal extent of the MOSAIC array. This delay corresponds to 2 frames at 50 Hz imaging. The present analysis was insensitive to such delays; though one could readily print voltage indicator islands on opposite sides of the array to measure conduction velocity and to attain sub-frame estimates of spike time at each island.

Capturing beating dynamics required a faster frame rate than used for HEK293 cell measurements, but recording at high speed under constant illumination during hour-long perfusion measurements was also not feasible due to phototoxicity and an overwhelming volume of video data. Figure 7a shows the implemented recording protocol: one frame with 490 nm and one frame with 405 nm illumination were recorded every 10 s, as for HEK293 cells. In addition, a 10 s, 50 Hz fluorescence recording was acquired every 2 min with 635 nm laser and 490 nm LED illumination to capture beating dynamics with voltage and calcium sensors (Fig. 7d). Simultaneous recording of the voltage sensor QuasAr in the far-red channel and GFP-based sensors in the green channel was possible through use of multicolor illumination and a multiband emission filter. Figure 7c shows images of the correlation between each pixel time-trace and the average 50 Hz recording time trace (with baseline trends removed) of the islands for several sensors. Pixels with high correlation have a clear fluorescence signature of beating, and thus map functional, sensor-expressing cardiomyocytes. Fluorescence correlation maps corresponded closely to mean fluorescence images, indicating that each island homogeneously expressed a single sensor. The uniform maps reveal good transduction efficiency across the island, and the sharp boundaries show minimal motion of the CM's after plating.

Recordings at baseline prior to administration of compounds (Fig. 7d), revealed synchronized APs (red) and cytosolic calcium transients (blue), as expected from cardiomyocytes with normal excitation-contraction coupling. We observed this behavior for multiple MOSAIC arrays (Supplementary Fig. 24). MOSAIC also provided simultaneous recording of calcium transients in the ER (purple) and mitochondria (green). The ER calcium concentration decreased as cytosolic calcium increased, indicating the presences of calcium-induced ER calcium release. Moreover, clear mitochondrial calcium oscillations were visible, synchronized with the APs and with the same sign as cytosolic calcium. Although both mitochondrial and cytosolic calcium levels increased quickly following depolarization, mitochondrial calcium returned to baseline levels more slowly than did cytosolic calcium. The slight phase lag of mitochondrial relative to cytosolic calcium suggests that mitochondrial calcium fluctuations were primarily driven by transport from the cytoplasm.

We used MOSAIC to study cardiomyocyte responses to selective inverse agonism, block and activation of β-adrenergic receptors (Fig. 7). CPG-20712A and ICI-118,551 are highly potent and selective for $\beta_1$- and $\beta_2$-adrenergic receptors, respectively (Table 2)—these compounds both suppress basal signaling (inverse agonism) and block effects of β-adrenergic agonists. They were applied alone and in combination with selective $\beta_2$ agonist formoterol and $\beta_1$ agonist dobutamine (Table 2). Finally, the co-agonist isoproterenol was applied. Administration of compounds is outlined in Fig. 7d, black text at bottom. Fluorescence was recorded from cells expressing the MOSAIC array over 60 min. Compounds were administered through a 10 s perfusion of the culture dish. Acute responses to the compounds were monitored immediately following perfusion and during the following 10-min period before the next intervention. Wash steps removed compound after each agonist addition (Fig. 7d); control buffer washes only modestly perturbed cardiomyocyte behavior (Supplementary Fig. 25).

Application of CGP-20712A resulted in a dynamic response in several of the arrayed sensors. Voltage recordings revealed a decrease in beat rate and the action potential broadening commonly observed with a longer interbeat interval (Fig. 7e), consistent with a reduction in basal $\beta_1$-adrenergic activity from an inverse agonist. At the same time, the amplitude of mitochondrial and ER calcium transients increased, as did the pH in all the cellular sub-compartments.

Subsequent $\beta_2$-adrenergic stimulation with 1 μM formoterol, still in the presence of $\beta_1$ block, resulted in an immediate increase in beat rate and decrease in AP width and amplitude. There were

**Table 2 β-adrenergic modulating compounds.**

| Compound | Function | $\beta_1$ $K_D$ (nM) | $\beta_2$ $K_D$ (nM) | $\beta_3$ $K_D$ (nM) | Refs. |
|---|---|---|---|---|---|
| CGP-20712A | Antagonist, inverse agonist | 1.5 | 780 | 6500 | 84–87 |
| ICI-118,551 | Antagonist, inverse agonist | 300 | 0.55 | 360 | 85–88 |
| Formoterol | Agonist | 780 | 2.3 | 1500 | 47,89,90 |
| Dobutamine | Agonist | 5900 | 1400 | 8100 | 46–48 |
| Isoproterenol | Agonist | 870 | 230 | 3000 | 47 |

Numerical data indicates binding coefficients, $K_D$.

also rapid reductions in the amplitudes of ER and mitochondrial calcium transients and in the pH of all cellular sub-compartments. The increase in cytosolic calcium transient amplitude, expected for positive inotropic $\beta_2$-adrenergic stimulation[41,42], did not occur immediately after perfusion but was observed in the subsequent high-speed recording 2 min later. This observation suggests a signaling cascade was required to activate the inotropic effect; likely mediated via cyclic AMP[43]. The increased cytosolic calcium transient amplitude was accompanied by a recovery in the AP amplitude. $\beta_2$ adrenergic agonism has been shown via experiments[44] and simulations[45] to increase AP amplitude via activation of $I_{Ks}$.

A wash, followed by application of ICI-118,551, reversed the effects of the formoterol as expected for a $\beta_2$-adrenergic inverse agonist. The beat rate and AP amplitude decreased to near-baseline values, and the AP50 spike width increased to near baseline. Cytosolic calcium amplitude decreased, and mitochondrial calcium amplitude increased. These effects were in concordance with the expected effects in voltage and cytosolic calcium dynamics following modulation of β-adrenergic receptors. MOSAIC further allowed us to detect fluctuations in the levels of ER and mitochondrial calcium and cellular pH that were previously not well studied.

In the presence of ICI-118,551, we then applied a high dose (100 μM) of dobutamine, which stimulated both α- and β-adrenergic signaling[46] and is reported to have a stronger maximal response in $\beta_1$ relative to $\beta_2$-adrenoreceptors[47]. Dobutamine is used clinically to increase heart contraction strength, but carries an associated arrhythmia risk[48]. Following application, the beat rate decreased slowly followed by intermittent or blocked beating, consistent with our previously observed results (Supplementary Fig. 26). The decrease in beat rate was accompanied by reductions in voltage and calcium amplitudes, opposite of what would be expected for a positive inotrope. Based on the response to formoterol, these hiPSC-CMs are highly sensitive to $\beta_2$-adrenergic signaling. Sustained suppression of basal $\beta_2$ signaling by ICI-118,551 may have overridden the expected effect of $\beta_1$ activation with dobutamine. This interpretation is consistent with the observation that $\beta_2$-adrenergic signaling predominates over $\beta_1$ in fetal cardiomyocytes and in hiPSC-CMs cultured for ≤30 days[49].

In addition to the reduced beat rate and reduced voltage and calcium amplitudes, the ICI-118,551/dobutamine treatment also caused AP triangulation, a widening of the AP base relative to its half max width, which has been associated with arrhythmia risk in vivo[50]. Unexpectedly, at the same time as the intermittent beating, synchronized, in-phase oscillations were observed in the mitochondrial and ER baseline calcium levels. The period of the oscillations, which is on the timescale of minutes, is much slower than calcium transients associated with action potentials. The cells maintained intermittent firing, AP triangulation, low

cytosolic calcium transient amplitudes, and oscillating ER and mitochondrial calcium levels through a wash and initial co-stimulation of $\beta_1$- and $\beta_2$-adrenergic receptors with isoproterenol. We inferred that the cells were still strongly perturbed by the extended suppression of basal $\beta_2$-adrenergic signaling by ICI-118,551 and the dobutamine treatment and had not yet returned to basal conditions after a wash. However, at ~4 min after isoproterenol addition, AP triangulation and cytosolic calcium transient amplitude began to return toward their initial values and baseline oscillation in mitochondrial and ER calcium ceased. By the end of the isoproterenol measurement and after a final wash, the cells had resumed regular beating, implying that effects of treatment with ICI-118,551 and high dose dobutamine were, at least partly, reversible.

**Discussion**

By simultaneously recording from many fluorescent sensors, MOSAIC enables study of interacting physiological parameters, an otherwise difficult task. In HEK293 cells, we observed correlated changes in pH, $Cl^-$, NADH, ATP, and cytosolic, mitochondrial, and ER calcium levels (Fig. 5). When the chemical energy source for the cells was perturbed, we observed initial perturbations to NADH and ATP, followed by returns to baseline as alternate metabolic pathways were activated. In contrast, cells could not recover when energy sources were removed and glycolysis and oxidative cellular respiration were pharmacologically blocked, despite the apparent increase in autophagic flux. MOSAIC revealed the initial coordinated calcium signaling in the mitochondria, cytosol, and ER in response to this strong perturbation, which has not previously been reported. The sub-second temporal correlations in such multimodal dynamic processes can help identify coupling between measured parameters. For example, the changes in chloride levels suggest an unanticipated involvement in these metabolic processes. Furthermore, the relative timing of signals informs on the direction of causality in signaling cascades. Changes in cytosolic calcium and ATP preceded changes in NADH and mitochondrial- and ER calcium, suggesting that cytosolic calcium and ATP may be upstream in the cascade of events.

MOSAIC also simultaneously tracked multiple physiological parameters in hiPSC-derived cardiomyocytes. Notably, action potentials and cytosolic-, mitochondrial-, and ER calcium transients were recorded simultaneously during spontaneous beating, with dips in ER-calcium levels and increases in cytosolic- and mitochondrial calcium for each beat, as expected. Beat rates, action potential amplitudes and calcium transient amplitudes were highly sensitive to manipulations of β-adrenergic signaling (Fig. 7). The amplitude of dips in ER calcium levels typically tracked the increase in cytosolic calcium transient amplitudes, consistent with direct flow of calcium from the ER to the cytosol. In contrast, the amplitude of cytosolic and mitochondrial calcium

transients often changed independently. For instance, after addition of CGP-20712A and formoterol, the cytosolic $Ca^{2+}$ amplitude initially decreased slightly and then recovered, while the mitochondrial $Ca^{2+}$ amplitude decreased substantially and only recovered very slowly. However, the mitochondrial $Ca^{2+}$ amplitude was inversely correlated with beat rate (Fig. 7e), which complicates interpretation. Another unexpected set of temporally correlated signals that would be difficult to detect without MOSAIC are the oscillations in baseline mitochondrial and ER-calcium levels during sporadic beating (Fig. 7e). Quantitative analysis of these effects will require additional replicates of these experiments and comparison of cardiomyocyte cell lines from different sources.

MOSAIC compares favorably to other approaches to recording from multiple sensors. One alternative approach is to record from sensors independently in separate wells. The single-well approach of MOSAIC permits sub-second cross-correlation of different modalities. Even with automated compound addition, it would be difficult to achieve this level of temporal precision across multiple wells. MOSAIC also decreases errors by ensuring that all measured cells have identical cell culture and pharmacological history and is more economical with precious cells such as iPSC-derived cardiomyocytes.

Another alternative approach is spectral multiplexing, where two or three sensors can be recorded simultaneously in well-separated fluorescence color bands. This approach has been taken by multiple groups for studying cardiomyocytes. Electrical pacing of the beat rate was paired with simultaneous imaging of voltage and cytosolic calcium[51] or voltage, cytosolic calcium, and contraction[52]. Similarly, optogenetic pacing, which was scaled to high-throughput applications, was paired with simultaneous imaging of voltage and cytosolic calcium[53] or voltage, cytosolic calcium, and contraction[54]. Spectral multiplexing is simpler to implement and can record from many cells per sensor but is limited to ~3 sensing modalities because of the broad widths of fluorescence absorption and emission spectra. Most sensors used in MOSAIC have only been developed in the GFP emission channel, so they cannot be spectrally multiplexed together. Whereas spectral multiplexing permits simultaneous measurement of fluorescent sensors in two or three spectral bands, MOSAIC enables measurements of tens of sensors simultaneously.

The highly parallel MOSAIC recordings facilitate discovery of unexpected relations between physiological parameters, which can contribute to hypothesis generation and follow-on targeted experiments to elucidate underlying mechanisms. The number of parameter pairs that can be examined for unexpected temporal correlations scales quadratically with the number of parameters measured, further motivating the high levels of multiplexing in MOSAIC.

An alternative approach to implement spatial multiplexing in MOSAIC is by patterning plasmids encoding fluorescent sensors instead of lentivirus[55]. DNA is transferred into the cells using reverse transfection, where plasmids are mixed with a transfection cocktail and printed as a microarray, and then cells are plated[56,57]. Using this approach, Kuchenov et al. recorded from 40 Förster Resonance Energy Transfer (FRET) fluorescent sensors to profile intracellular signaling network activity[55]. Transfected cell spots were imaged serially with 3 min temporal resolution, much slower than the <10 ms temporal resolution possible with the Firefly microscope. Microarrays based on chemical transfection of DNA plasmids may be preferred for experiments in easily transfected cell lines such as HEK293 because virus need not be produced and the printed DNA microarrays can be stably stored for more than a year before cell plating[57]. The MicroSCALE platform for lentivirally-based cellular microarrays was developed for difficult-to transfect cell lines and for experiments where

reporter expression must last longer than ~3 days[23]. For cardiomyocytes, which are difficult to transfect and take days to fully express fluorescent sensors, lentiviral microarrays are preferred.

The MOSAIC platform yields rich, multimodal data, but its implementation can be challenging. The hardware requirements include a microarray printer, a wide field-of-view, high-speed microscope like the Firefly[24], and a perfusions system. Production of lentiviruses with a consistently high titer for each sensor is labor intensive. Preparation of the MOSAIC arrays is logistically complex because freeze/thaw cycles degrade viral infectious titer; we printed lentiviral arrays and plated cells on the same day. Another complexity is that each sensor must be handled appropriately as they have different sensitivities, fluorescence excitation and emission requirements, analysis methods, pH sensitivity, and concentration ranges over which they perform well. MOSAIC is limited by the performance of the included fluorescent sensors, and in many cases quantifying absolute analyte concentrations is difficult. An inherent limitation of the approach is that sensors are expressed in different cells. If there is large heterogeneity between cells in their dynamic responses, correlations based on population averages might miss important features of the data.

Looking forward, the MOSAIC platform will grow as new and improved fluorescent sensors become available. For example, an improved pH ratiometric sensor[58], more sensitive glucose sensor[59], and ATP sensors targeted to the ER and mitochondria[60] are anticipated enhancements. A natural future direction would be to add additional reporters of second messengers such as cyclic AMP (e.g., ref. [10]) or $PIP_2$ (e.g., ref. [12]). With multiple second messengers and the calcium sensors targeted to different subcellular locations, one could measure multiple key signaling pathways simultaneously. By selectively stimulating different GPCRs, one could start to unravel complex signaling cascades.

Another potential application is in disease modeling. One could prepare MOSAIC arrays and plate iPSC-CM's containing disease-associated mutations on half the arrays and WT iPSC-CM's on the remaining arrays. One could then challenge the cardiomyocytes with, for example, metabolic stress or adrenergic stimulation, and look for different responses between mutant and control cells. MOSAIC data could provide critical clues for discovering pathways impacted by the disease. A similar methodology could be applied to unraveling the mechanisms of action for small molecules in a drug discovery pipeline.

A different strategy would be to combine the array of fluorescent sensors with gene editing technologies. In CRISPR interference (CRISPRi)[61], for example, guide RNAs can be encoded in lentiviral plasmids to silence target genes. One could knockdown one gene in the entire array using CRISPRi and look at the effect of knockdown on the full set or reporters, or one could knockdown a different gene in each island in the array and look at the effect with one reporter. With many genes and many reporters, it is easy to imagine a very large number of simultaneous measurements, where the efficient use of cellular reagents with MOSAIC would be critical. In summary, MOSAIC is a versatile and information rich technology with a wealth of opportunities for studying complex cellular processes.

## Methods

**Cloning the MOSAIC constructs**. Fluorescent sensors were collected from many labs and each had different restriction sites. To address the diversity in starting constructs, we used Gateway® cloning[62]. In addition to the lentiviral backbone used for MOSAIC, the Entry vector for each sensor facilitates any future cloning into different backbone plasmids. We first PCR amplified each insert, adding the Gateway attB1 and attB2 sequences to the 5′ and 3′ end of the open reading frame (ORF). Using BP Clonase II (ThermoFisher #11789100), we inserted the ORF into the pDONR221 plasmid (ThermoFisher #12536017) to create the Entry plasmid. Using LR Clonase II Plus (ThermoFisher # 12538120), we then transferred the ORF

into the pLX304 Gateway Destination vector, a 3rd generation lentiviral backbone used to make a genome-scale lentiviral library of human ORFs[63].

**Lentivirus production.** Lentivirus was prepared in HEK293 cells following established protocols[39,64]. Briefly, HEK293 cells with a low passage number were cultured in gelatin-coated (Stemcell Technologies, #07903) 15cm dishes in DMEM + 10% FBS. When cells reached 80% confluence, they were transfected in serum-free medium with polyethylenimine (PEI; Sigma 408727). To transfect, we first mixed the plasmids (14 µg of the vector plasmid, 9 µg of the 2nd-generation packaging plasmid psPAX2 (Addgene #12260), and 4 µg of viral entry protein VSV-G plasmid pMD2.G (Addgene #12259)) in 1.2 mL of DMEM. We then added 36 µL of 1 mg/mL PEI, vortexed, incubated for 10 min for complexation, and added drops distributed around the 15cm dish. Four hours after transfection, we aspirated the toxic PEI solution and replaced with 16 mL DMEM + 10% FBS. We harvested the supernatant 48 h after media exchange, centrifuged 5 min at $500 \times g$ to pellet cellular debris, and filtered the supernatant with a 0.45 µm filter.

**Concentration of lentiviral particles.** High-titer lentivirus stocks were prepared through ultracentrifugation of harvested lentivirus-containing cell culture medium. Briefly, low-titer lentiviral supernatants were layered on top of 20% sucrose cushions in ultra-clear ultracentrifuge tubes (Beckman 344058), and transferred to a SW-28 rotor (Beckman). Samples were ultracentrifuged at $126,000 \times g$ for 2 h at +4 °C, and supernatants were discarded. Pelleted virions were resuspended in 100 µL printing buffer[23] (0.4 M HEPES, 1.23 M KCl, trehalose (12.5 mg/mL), and protamine sulfate (12 mg/mL), with pH adjusted to 7.3), aliquoted, and either used immediately or transferred to −80 °C until further use.

**Substrate preparation for printing.** To print patterned arrays, we prepared a chemically activated substrate in glass-bottomed dishes (Cellvis #D35-20-1.5-N) as described in ref. [39]. In brief, we first covalently bonded a polyacrylamide (pAA) gel to the glass surface using silane chemistry. The pAA is very cyto-repellant: no cells adhered to an un-adorned pAA surface. The gel thickness was set to ~40 µm, and the stiffness, controlled by the amount of bis-acrylamide crosslinker, was set to ~20 kPa[65,66]. The polyacrylamide was activated to covalently bind primary amines in the lysine side chains of fibronectin by doping the pAA gel with N-hydroxysuccinimide (NHS) leaving groups. Chemically activated plates could be vacuum sealed under nitrogen and stored for months at −80 °C.

To prevent migration of motile cells such as HEK293 cells, we printed fibronectin (Yo Proteins #663) into islands using the microarray printer. For non-motile cells such as cardiomyocytes, we coated the entire surface of the pAA gel with fibronectin (50 µg/mL) for 30 min. at room temperature. In preparing fibronectin surface coatings one must take care to ensure the Tris or other buffers containing primary amines are rigorously excluded from the solution as they will react with the NHS groups.

**Microarray printing.** Microarray printing was performed using a Gene Machines OmniGrid equipped with MicroQuill pins (Major Precision). The chemically activated dish was stuck to a 1 × 3 inch microscope slide using modeling clay, and the slide was mounted in the microarray printer. High-titer lentivirus and fibronectin reagents were prepared in a conical bottom 384-well inking plates (Molecular Devices #X6004) to minimize reagent usage. We varied fibronectin concentration and solution composition and settled on the optimal 200 µg/mL in PBS and 1% glycerol. Forty microliters of the fibronectin solution was added to one well of the inking plate. After ultracentrifugation, the high-titer lentivirus was resuspended in viral printing buffer (HEPES (0.4 M), KCl (1.23 M), trehalose (12.5 mg/mL), and protamine sulfate (12 mg/mL), pH adjusted to 7.3) following ref. [23]. Ten microliters of each virus was loaded into different wells of the inking plate.

Viral and FN printed spots were about 120 µm diameter on the polyacrylamide surface. Cell islands were made by printing 3 × 3 arrays of spots at a 100 µm pitch to make roughly 340 µm square islands. The islands had a 500 µm center to center spacing. Printing parameters were: dipping time in fibronectin/virus solution: 2.5 s; printing contact time: 100 ms; print pin acceleration/deceleration at surface: 150 cm²/s. The contact time and acceleration/deceleration had only a minor impact on spot size and quality. Between printing solutions, the pin was cleaned by 3 sonication and vacuum drying cycles. To increase the number of virions on each island, we printed the same full pattern twice, giving the virus time to dry in between. The patterned viral arrays were stored on ice, and cells were plated on the array the same day they were printed.

**Cardiomyocyte culture and imaging.** Human induced pluripotent stem cell-derived cardiomyocytes (hiPS-CM; iCell cardiomyocytes), plating media, and maintenance medium were purchased from Cellular Dynamics International. Cells were thawed and cultured following manufacturer instructions and maintained at 37 °C with 5% CO₂. Cells were plated at a density of 50 k/cm², calibrated by a Trypan blue stain and accounting for the lot plating efficiency specified by the manufacturer. Cells were plated in just the recessed region of the glass-bottomed dish onto the freshly printed MOSAIC arrays and allowed to attach for 1 h at room temperature. After 1 h, 1.5 mL of additional plating media was added, and cells were transferred to the incubator. Forty-eight hours after plating, the plating

medium was aspirated and replaced with maintenance medium. The medium was exchanged every other day after plating.

Cells were imaged after 10–14 days post plating. Prior to imaging, the medium was replaced with a low-autofluorescence imaging buffer (in mM: 1.8 CaCl₂, 2.5 × 10⁻⁴ Fe(NO₃)₃, 0.81 MgSO₄, 5.3 KCl, 129 NaCl, 0.91 NaH₂PO₄, 1 sodium pyruvate, 10 D-(+)-galactose, 25 HEPES). The imaging buffer matched the osmolarity and ion concentrations of the maintenance medium, but lacked phenol red, vitamins, and amino acids and was buffered by HEPES to minimize pH changes. After changing to the imaging media, cells were allowed to equilibrate for 30–60 min (37 °C, ambient CO₂) prior to imaging on the microscope at 37 °C.

Functional imaging was performed with the quadband emission filter Semrock #FF01-446/510/581/703-50 to enable recording from sensors in multiple color bands simultaneously. The 473 nm illumination intensity for CheRiff stimulation during cardiomyocyte pacing was 100–150 mW/cm², the 635 nm illumination intensity for QuasAr voltage imaging was 20 W/cm², the 485 nm illumination intensity for calcium imaging was 60 mW/cm², and intensities at longer exposure times for slower sensors were lower. To capture dynamics at different time scales, high frame-rate recording was interleaved with regularly recorded frames: 10 s of 50 Hz fluorescence recording was acquired every 2 min with 635 nm laser and 490 nm LED illumination to capture beating dynamics with voltage and calcium sensors. Every 10 s throughout the 60 min recording, one frame with 490 nm and one frame with 405 nm illumination was recorded.

**HEK293 culture and imaging.** Low passage-number HEK293 cells (ATCC #CRL-3216) were plated onto freshly printed MOSAIC arrays at 40% confluence. Cells were allowed to adhere to the surface for 30 min at room temperature before transfer to the incubator (37 °C, 5% CO₂). We found that rinsing the MOSAIC arrays before cell plating or after HEK293 cells had been allowed to adhere increased cross-contamination between islands. Cells were cultured in DMEM + 10% FBS at 37 °C and 5% CO₂.

HEK293 cells were imaged at 2–3 days post plating, by which time the islands displayed sensor fluorescence. By 4 days, the HEK293 cells grew into a 3-dimensional mound of cells which would easily lift off the dish. Imaging buffer and cell handling, and microscope conditions were identical to cardiomyocytes.

**Statistics and reproducibility.** Micrographs showing expression and trafficking of individual sensors in HEK293 and cardiomyocytes were recorded on either a commercial inverted fluorescence microscope, a spinning disk confocal microscope, or the Firefly microscope (Figs. 2a, c, 3g–n, 6c–f and Supplementary Figs. 2–22). Typically, 1–3 images were captured from different fields of view and a representative image was selected for publication. Images of HEK293 cell MOSAIC arrays (Fig. 3a–f, o) were drawn from ~11 rounds of measurement. The selected images were from a representative round; it was not uncommon to observe an overgrown or missing island in an array. Images of cardiomyocyte MOSAIC arrays (Figs. 6a, b, 7b, and Supplementary Fig. 24) were representative of ~11 rounds of measurement. Functional recordings (Figs. 2b, d–f, 5, 7d, e and Supplementary Figs. 2, 4–22) were typically drawn from a single measurement in one culture dish. Multiple robust effects were observed in multiple rounds, including cardiomyocyte action potentials and calcium transients (Figs. 2e, 7d and Supplementary Figs. 2c, 5g, 6f, 24, 25a, 26), responses to pH changes (Fig. 4 and Supplementary Figs. 8–13), and responses to redox changes (Figs. 2b, d, 5 and Supplementary Figs. 14–18).

**Microscope and automation.** Experiments were conducted on a home-built inverted fluorescence Firefly microscope[24] (Supplementary Fig. 1). Built around the Olympus 2x MVX Plan Apochromat objective (numerical aperture 0.5), the microscope has a large field of view (FOV) and high light collection efficiency. Fluorescence is imaged onto a scientific CMOS camera (Hamamatsu Orca Flash 4.0), resulting in 3.25 µm spatial resolution, a 6 × 6 mm FOV, and a 100 Hz maximum frame rate for the full FOV. The microscope has three classes of illuminators. (1) For voltage imaging with QuasAr[9], high-power 635 nm laser light was coupled into the sample with a custom fused silica prism just below the critical angle for near-total internal reflection (TIR). As in light-sheet microscopy, the near-TIR geometry minimizes background autofluorescence and increases illumination intensity via the refractive beam compression at the glass-water interface. (2) To target optogenetic stimulation to specific cells or MOSAIC islands, patterned 405 and 473 nm laser illumination was reflected off a digital micromirror device (DMD, Digital Light Innovations DLi4130 – ALP HS). The binary pattern could be refreshed (triggerable) at over 20 kHz with a spatial resolution in the sample of 7 µm. (3) Additional illumination colors for various MOSAIC sensors were provided by a six-LED system mounted above the sample (Thorlabs #M365L2, M405L2, M455L3, M490L3, M530L3, M565L3). Each LED was collimated with an aspheric lens and spectrally windowed with an excitation filter before being combined by dichroic mirrors. Each LED had independent, high-speed analog control (Thorlabs # DC4104). Excitation filters included Semrock #FF01-448/20-25, FF02-529/24-25, FF01-565/24-25 and Chroma # ET365/10x, ET405/20x, ET485/20x, and dichroic mirrors included Chroma #T387lp, T425lpxr, T470lpxr, T510lpxr, T550LPXR. The LEDs were first imaged onto an iris (not shown in Supplementary Fig. 1) to limit the illumination field, and the iris was imaged onto the sample. LED light was

incident on the sample at ~45° to minimize the amount of light passing through the objective and the associated autofluorescence.

All equipment was synchronized with a data acquisition (DAQ) card (National Instruments #PCIe-6353) and a custom LabVIEW program. The camera was run in synchronous mode with a rolling shutter, so the start and stop of every frame was triggered with a digital signal. A waveform was loaded onto the DAQ card for each light source, valve (see perfusion below), DMD trigger, and camera trigger, so all equipment was synchronized with microsecond temporal resolution. Because many of the ratiometric MOSAIC sensors required multiple illumination wavelengths, typical recordings involved capturing a frame (100ms exposure) with 490nm illumination, followed immediately by a frame (100ms exposure) with 405nm illumination. Depending on sensory dynamics, typically 5–10 s elapsed before the next pair of frames was recorded. When operating with multiple colors and a rolling shutter in the camera's synchronous mode, every other frame (when lights are switching on and off) is bad and was discarded during analysis.

The microscope had sample temperature control and recordings were made at 37 °C. The microscope was also equipped with gas control over the sample, but we found HEPES buffering with ambient atmospheric gas was more robust than carbonate buffering and 5% $CO_2$.

**Perfusion**. A perfusion system (Supplementary Fig. 1) provided rapid buffer exchange for pharmacological perturbations. To test each fluorescent sensor, typically a different buffer was required. 30 mL syringes loaded with media were mounted in a home-built aluminum heater block maintained at 37 °C. Each line had a pinch valve under digital control (Warner #VC-8P) and was combined into a common path by a manifold. The flow rate was adjusted with a metering valve (Swagelok #SS-SS1-VH). Prior to flow into the sample chamber, we equipped the system with a switching valve that could send the flow either to the sample or to waste. To minimize the dead volume in the perfusion lines and thus media exchange time, we washed the lines through the switching valve with media from the desired syringe before flow was routed to the sample. The buffer temperature was maintained with insulation and a resistively heated wire before flowing into the sample chamber.

A perfusion chamber was implemented in the glass-bottomed dishes (Supplementary Fig. 1C) by inserting a 30mm diameter fused silica disc (McMaster #47-834) with two holes ground in it (Supplementary Fig. 1B). We bonded Pyrex cylinders (VWR #22877-254) over each hole to with UV-cure adhesive to increase the functional volume before overflow. The perfusion system dripped into one hole, and suction removed excess liquid, providing flow through the 1mm gap between the cover glass and the fused silica disc (Supplementary Fig. 1D). A high-speed recording of fluorescent beads perfused through the system revealed quasi-laminar and complete volume exchange of the area over the MOSAIC array in around 1 s. By making the entrance hole oversized, bubbles from the perfusion line typically floated to the surface and did not enter the chamber. We used a flow rate of 5 mL/min and exchanged the chamber volume >3x for a total buffer exchange time of 15 s.

**Calculation of fluorescence time traces**. Using a custom MATLAB program, we entered the array dimensions and the selected the centers of the four corner islands manually. Using the well-defined island sizes and inter-island spacings patterned by the microarray printer, the analysis program selected tight bounding boxes around each island. The time-dependent background is estimated by the average fluorescence in a frame encircling each island and subtracted. For most islands, the flat-average of the tight bounding box gave an accurate sensor response, but there was occasional cross-contamination where a few cells expressed the sensor from a neighboring island. If the contaminating cell(s) expressed a very bright or responsive sensor, it could significantly change the average trace, so small sub-regions had to be excluded when calculating the island time traces. Each island was sub-divided into 16 sub-regions, the average time-trace for each was calculated, and the zero-time cross-correlations between sub-regions were calculated. Before calculating cross-correlations, the mean and linear components were subtracted from each time trace to minimize effects of photobleaching and baseline drift, and each time trace was normalized so its zero-time autocorrelation was 1. Regions were ranked by their average cross-correlation; by default, the lowest 25% were discarded. For all data in this paper, the automated sub-region keep/discard calls were manually curated to ensure that sub-regions displaying any unusual signal or containing unusually bright cells were appropriately discarded. A simple graphical user interface (GUI) streamlined this process; ~5 min were required to curate the array. As we accumulate MOSAIC recordings and build confidence in thresholds and outlier detection algorithms, the manual curation step can likely be eliminated in the future.

A mask was generated that discarded the contaminated regions, and we used this mask to calculate the average time trace. To increase the signal-to-noise ratio, get a more accurate measure of fractional fluorescence change, and further suppress the contribution from other sensors, pixel weighting was used in the unmasked regions of each island. To determine weights, the time-trace for each unmasked pixel was correlated with the average fluorescence trace, and pixels with strong correlation and a large $\Delta F/F_0$ were more heavily weighted when re-calculating the average, as in ref. [67]. The pixel weighting mitigated the effects of bright fluorescent contaminants that did not respond as other cells in the island and mitigated the effects of trace contaminants that were not masked out. The

same mask was applied to movies recorded using 405 and 488nm excitation, and the paired time traces were used to determine the final sensor readout as described in the "Calculated quantity" column in Table 1.

**pH correction**. Many of the MOSAIC sensors are sensitive to pH, and pH-induced changes must be removed to reveal the primary physiological responses the sensors are meant to detect. The pH response of each sensor was calibrated initially by changing the buffer pH stepwise from 6.5 to 7.75 (Fig. 4). To equilibrate the pH of the cytosol and subcellular compartments with the buffer pH, we added the $K^+/H^+$ exchanger nigericin at 14 μM. To prevent a $[K^+]$ gradient from driving a proton gradient, we used a high-potassium buffer[6,68]. The buffer composition was (in mM): Good's zwitterionic buffer 25, KCl 100, NaCl 38, $CaCl_2$ 1.8, $MgSO_4$ 0.8, $NaH_2PO_4$ 0.9. The Good buffer, chosen based on its pKa and effective buffering pH range, was MES for pH 6.5 and HEPES for pH 6.75–7.75.

After perfusion of the buffers with different pH values, we waited 1 min for the pH to equilibrate in all the cellular sub-compartments and recorded the steady-state fluorescence for each island. The sensor response $S$ was tabulated, calculated as described in the "Calculation of fluorescence time traces" section and Table 1. We recorded pH titrations on four MOSAIC arrays to build statistics. The results of the pH titration were fit to a Hill equation:

$$S = S_{base} + \frac{S_{acid} - S_{base}}{1 + \left(\frac{[H^+]_{50}}{[H^+]}\right)^\gamma} \qquad (1)$$

$S_{base}$ is the fully deprotonated sensor signal, $S_{acid}$ is the fully protonated sensor signal, $[H^+]_{50}$ is the proton concentration where the sensor signal is at half brightness (corresponding to pH = $pK_a$), and $\gamma$ is the Hill coefficient, which determines the slope of the curve. Example titration data and fits are shown in Fig. 4b. The coefficients from the fit are included in Supplementary Table 1.

For each subsequent MOSAIC recording, the time varying $pH(t)$ was calculated for each subcellular location from the pHluorin pH sensor by inverting the polynomial fit and applying it to the pHluorin time trace:

$$[H^+](t) = [H^+]_{50} \left(\frac{S_{base} - S(t)}{S(t) - S_{acid}}\right)^{1/\gamma} \qquad (2)$$

To calculate pH, we used the bright and sensitive superecliptic pHluorin[6,69] because we had difficulty getting reliable readouts from the original ratiometric pHluorins[6,69], which are dimmer and less sensitive. Using the extracted $pH(t)$ for each cellular subcompartment, we calculated each sensor time trace expected to result from pH changes alone by applying the Hill equation with the parameters from Supplementary Table 1. The predicted pH response was subtracted from the raw sensor response to yield the final time trace.

The deterministic correction described above is imperfect for two reasons: (1) for each island there may be a small amount of background autofluorescence, and (2) there may be some variation in the starting intracellular pH despite the fact that cells are always prepared following the same protocol. The background is negligible for most bright sensors since we subtract the background from a frame encircling each island (see the section "Calculation of fluorescence time traces"), but it can be critical for sensors such as superecliptic pHluorin, which becomes quite dim at acidic pH. These two uncertainties manifest as an offset and scaling in the original Hill expression:

$$S = a\left(S_{base} + \frac{S_{acid} - S_{base}}{1 + \left(\frac{[H^+]_{50}}{[H^+]}\right)^\gamma}\right) + b \qquad (3)$$

$a$ captures the uncertainty in starting pH and $b$ captures the background offset. We used $a$ and $b$ as fitting parameters for pH correction. For each sensor, $a$ was constrained to the range [0.5, 2] and $b$ was constrained to the range [−0.05, 0.05].

We did not apply pH correction to the roGFP-based redox sensors (pQS012–pQS016) and the NADH sensor Peredox (pQS023). The ratiometric roGFP sensors showed fluorescent deflections in the same direction for both excitation wavelengths in response to pH, but in different directions for changes in redox state, so the pH artifact in the ratio was small. Peredox was also insensitive to pH changes, as it was evolved to minimize the pH response[28]. An additional reason for not correcting the pH response of these sensors is that their pH response was non-monotonic, perhaps due to changes in redox state or NADH/NAD$^+$ ratio during the pH titration protocol.

Reliably fitting and removing the pH response is not trivial, because changes in the physiological parameter reported by each sensor may be correlated with changes in pH. To reduce the complexities associated with pH correction, there are two key improvements that could be implemented in the future. First, the superecliptic pHluorins can be replaced by a brighter ratiometric pHluorin[58], aiding in quantification of the pH in each cellular compartment. Second, a pair of pH steps, with membrane permeabilization with nigericin, can be added to the end of each perfusion protocol to force the cells to known intracellular pH values. These steps can be used to reliably calibrate the pH response of each sensor. With these two advances, the fitting parameters $a$ and $b$ would not be needed.

**Autophagic flux assay and western blot**. HEK293 cells were pretreated with 100 nM Bafilomycin A1 (Sigma-Aldrich) for 2 h to inhibit lysosome acidification and

autophagosome/lysosome fusion. Media was exchanged to treatments indicated in Supplementary Fig. 23 and cells were incubated for 0, 1, or 4 h, maintaining Bafilomycin concentration. Autophagic flux was assessed by western blotting for accumulation of autophagosome marker LC3-II. Cells were scraped off the plate in ice-cold PBS, pelleted at +4 °C, and lysed in RIPA buffer (50 mM Tris-HCl pH 8.0, 150 mM NaCl, 1% NP-40, 0.5% SDS, Sigma-Aldrich) supplemented with protease and phosphatase inhibitors (Roche). Lysates were cleared by centrifugation and equivalent amounts of protein (30 μg) were subjected to SDS-PAGE under reducing conditions on Bis-Tris gels (Invitrogen). Proteins were transferred to nitro-cellulose membranes (BioRad) and detected using primary antibodies raised against LC3A/B (Cell Signaling Technology, #12741) and GAPDH (Sigma-Aldrich, G9545), an HRP-conjugated goat anti-rabbit secondary antibody (Abcam ab6721), and enhanced chemiluminescence substrate (BioRad).

**Reporting summary**. Further information on research design is available in the Nature Research Reporting Summary linked to this article.

## Data availability

Data are available from the corresponding author upon reasonable request. The source data underlying Supplementary Fig. 23a are provided as a Source data file. Source data are provided with this paper.

## Code availability

Data analysis codes are available from the corresponding author upon reasonable request. Source data are provided with this paper.

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

## Acknowledgements

We thank Katherine Williams for extensive help cloning the constructs and Jun-Han Su for help with MOSAIC recordings. We thank many labs and individuals for providing the fluorescent sensors: Dr. Julia Birk and Prof. Christian Appenzeller-Herzog for Grx1-roGFP1-iE_ER; Prof. Tobias Dick for Grx1-roGFP2, mito-roGFP2-Grx1, roGFP2-Orp1, and mito-roGFP2-Orp1; Prof. Gary Yellen (via Addgene) for Peredox and PercevalHR; Prof. Wolf Frommer (via Addgene) for FLII12Pglu-700μδ6; Prof. Colin Akerman (via Addgene) for ClopHensorN; Montana Molecular for Upward DAG; Dr. Douglas Kim for GCaMP6F; Dr. Mark Henderson and Dr. Brandon Harvey for GCaMPer; Dr. Sarah Kettlewell and Prof. Godfrey Smith for Mitycam; and Prof. Gero Miesenbock for ratiometric and supereclictic pHluorin. This work was supported by the Howard Hughes Medical Institute (C.A.W., S.B., and A.E.C.), a Vannevar Bush Faculty Fellowship (C.A.W. and A.E.C.), and the Wenner-Gren Foundation (A.R. and E.M.H.).

## Author contributions

C.A.W. contributed to experiment conception, design, and execution, analysis, and writing. S.B. contributed to experiment design and execution, analysis, and writing. A.R. performed the autophagic flux assay. E.M.H. contributed to experiment design, execution, and writing. A.E.C. contributed to experiment conception and design and writing.

## Competing interests

The authors declare no competing interests.
