## [Peer Review File · Nature Communications]

Reviewers' Comments:

Reviewer #1:

Remarks to the Author:

I enjoyed reading this manuscript, which describes a smart technical approach to simultaneously apply different genetically encoded probes in a microarray assay. The authors used defined lentivirus particles that were blotted on a microarray to infect cells and consequently express different genetically encoded probes in cells of interest. A special wide-field microscope was used to image the grouped fluorescent probes in HEK293 cells or cardiac myocytes simultaneously. The novel technology was named MOSAIC, which seems an appropriate name. The sophisticated technology was well described and its potential was nicely demonstrated.

The following corrections are necessary prior to publication of the work:

On page 4 it should be "Figure 2e" not "Figure 3e" and "Figure 2f" and not "Figure 3f".

I have these concerns/suggestions:

1.) The Ca²⁺ measurements shown in Figure 5 are less convincing. Why at the beginning of the protocol Ca²⁺ in the ER and cytosol is decreasing while mitochondrial Ca²⁺ seems to increase stepwise? If under these conditions of energy stress, Ca²⁺ is mobilized from the ER, Ca²⁺ should consequently first increase in the cytosol (which is not the case) and then within mitochondria. It is also difficult to understand why Ca²⁺ increases transiently within the cytosol and ER during addition of rotenone and oligomycin. During this phase of the protocol, Ca²⁺ is restored within depolarized mitochondria. The authors should give a plausible explanation for these subcellular Ca²⁺ fluxes. In addition it might help to study and compare well-known Ca²⁺ signals in response to an activation of e.g. the IP₃ pathway using ATP. Ca²⁺ signals between the sub-compartments (cytosol, ER, Mitochondria) from the cell population might not correlate well due to cell-to-cell variations. Hence, some comparisons with single cell responses might be helpful.

2.) Also in Figure 5: Why is the glucose level not reduced when glucose is removed from cells? Was glucose already removed prior to imaging?

3.) Measurement of mitochondrial ATP dynamics might represent a more informative response compared to cytosolic ATP changes; see Depaoli et. al Cell Reports 2018; DOI: 10.1016/j.celrep.2018.09.027. Also for ATP changes huge cell-to-cell variations have been shown. The authors might better discuss that the low resolution (2 x objective) does not well allow to resolve single cell responses.

4.) The authors should cite Kuchenov et al., 2016, Cell Chemical Biology; <http://dx.doi.org/10.1016/j.chembiol.2016.11.008>. This work describes a similar approach using plasmids and FRET-based genetically encoded probes.

Reviewer #2:

Remarks to the Author:

The authors have developed an impressive tool to parallelize a large number of optical measurements in a single system. The technique might be very useful for the field and is thoroughly characterized. Internal correction of the probe signals for pH effects is elegant, highly valuable and makes the setup robust.

While the technique itself is impressive and suitable for the journal, the biological relevance and results are not yet convincing and require some extra work. hPSC-CM experiments seem minimal. Furthermore, it should be made clear whether this tool is 'just' an all-in-one tool to measure multiple parameters or whether there is a benefit in measuring those parameters simultaneously – as is claimed.

Major points:

- Advantage 4 ("For cells that communicate over long distances via electrical signaling such as a cardiomyocytes or neurons, behavior can be synchronized throughout the well.") does not take into account propagation times required for signals to travel. While this might not be problematic for slow response behaviour this will limit time-correlation between fast-response transients such

as AP, Ca and contractility. In turn, this might also provide an opportunity: is it possible to assess propagation, a key parameter for excitable cells, using this system?

- Details are required on the efficiency of transfection, especially when using hPSC-CMs since they are known to be not easily transfected. The manuscript should include transfection efficiency numbers so the number of cells from which the signals originate can be assessed. For example: cardiomyocytes are quite large so according to the calculations, islands might contain tens of cells. If the transfection efficiency is below 20% - which might not be an unrealistic estimate based on sensor 20 in figure 7b for example - this would result in maybe 30 cells * 20 percent = 6 cells. Since hPSC-CM populations are not entirely pure, supporting cell types such as fibroblasts might be the ones transfected instead of cardiomyocytes. More quantitative details on the transfection efficiency and identity of the target cells are required.
- Importantly, the results from the measurements with cardiomyocytes seem to be a single experiment and results are not quantitatively analysed. As standard in the field, this experiment should be repeated at least three times and claims such as changes in triangulation, oscillations, amplitudes, etc. should be made based on quantitative data from those repeated experiments. In addition, it should be corrected for baseline/vehicle measurements (wash in, wash out of buffers, etc.).
- In line with the previous point, the authors mention "The increase in cytosolic calcium transient amplitude, expected for positive inotropic β 2-adrenergic stimulation⁷², did not occur immediately after perfusion but was observed in the subsequent high-speed recording 2 min later, implying that a signaling cascade was required to activate the inotropic effect." but also such an increase in AP amplitude is seen which is unexpected. How can the reader be sure this increase in amplitude is not a washing artefact? Again, statistical comparison to baseline changes from multiple experiments are needed before making such claims.
- In the discussion, the authors need to be more critically reflective on how their system compares to others. For example, how does it compare to other integrated optical solutions (For example Clyde Bioscience System and <https://www.nature.com/articles/s41467-019-12354-8>)? What conclusions can be drawn that cannot be drawn if the data are not measured simultaneously? Or is it merely increased throughput? What are the limitations of the system? Also, the analysis seems to require a lot of manual tweaking because of cross contamination. How does this affect the throughput?
- Figure 7b, in 11 there seems to be a labelled region/area (or contamination??) stretching from one island to another one. What is happening here?

Minor points:

- Figure 2e and 2f are described in the text as 3e and 3f.
- In figure 7d, please add the names of the compounds. It is now a bit hard to follow.

Reviewer #3:

Remarks to the Author:

To clarify simultaneously the different response behaviors in cells treated by chemical or physical treatment, the authors of this manuscript developed a technique, Multiplexed Optical Sensors in Arrayed Islands of Cells (MOSAIC). This method enables parallel sensing for different signaling or metabolism species or events utilizing the patterned cells encoded with different fluorescent sensor with microarray printing the sensor encoding lentiviral vectors. The special design for motile and nonmotile cells has also been displayed. This technique is a new advance in synchronous sensing of cell events in a number of living cells, which provides more accurate same culture conditions and timing sequence. It is desirable the rapid publication in this journal. Before the acceptance for publication, the following concerns should be addressed.

1. The marker detection for the proposed autophagy in HEK293 cells upon nutrient deprivation should be carried out to confirm the hypothesis. Otherwise, the related discussion is questionable.
2. What is exact reason for the different plasmid number in different islands, especially the islands

for same sensing target? And how to guarantee the desired number in practical experiments.

3. The immobilization of motile cells might be discussed further, how to evaluate the effect of immobilization and is there any other measures to improve the immobilization of the cells?

Response to reviewer comments

We thank the Reviewers for their careful reading of the manuscript, overall positive comments, and constructive critiques. Below reviewer comments are in black and responses are in blue.

Reviewer #1 (Remarks to the Author):

I enjoyed reading this manuscript, which describes a smart technical approach to simultaneously apply different genetically encoded probes in a microarray assay. The authors used defined lentivirus particles that were blotted on a microarray to infect cells and consequently express different genetically encoded probes in cells of interest. A special wide-field microscope was used to image the grouped fluorescent probes in HEK293 cells or cardiac myocytes simultaneously. The novel technology was named MOSAIC, which seems an appropriate name. The sophisticated technology was well described and its potential was nicely demonstrated.

We thank the Reviewer for the positive assessment.

The following corrections are necessary prior to publication of the work:

On page 4 it should be “Figure 2e” not “Figure 3e” and “Figure 2f” and not “Figure 3f”.

Corrected.

I have these concerns/suggestions:

1.) The Ca²⁺ measurements shown in Figure 5 are less convincing. Why at the beginning of the protocol Ca²⁺ in the ER and cytosol is decreasing while mitochondrial Ca²⁺ seems to increase stepwise? If under these conditions of energy stress, Ca²⁺ is mobilized from the ER, Ca²⁺ should consequently first increase in the cytosol (which is not the case) and then within mitochondria.

The physiological state of the cells could not be precisely controlled as the cells were brought from the incubator to the microscope. We exchanged to imaging buffer before placing cells on the microscope, but found marked transients in many of the sensors during the first buffer exchanges. To minimize the impact of these transients, we began all experiments with two washes with imaging buffer, bringing the cells to better-controlled initial conditions. We did not try to interpret initial transients during these washes. We added a sentence to the manuscript at the beginning of the figure 5 description.

We do not think that a substantial cytoplasmic Ca²⁺ increase must precede the mitochondrial Ca²⁺ increase. See e.g. Ref. ¹, which shows that there can be ER-to-mitochondria Ca²⁺ interchange with little change in the cytoplasm. Compare, however, to recent work showing that in neurons direct cytoplasm-to-mitochondria Ca²⁺ uptake is much more efficient than in other cell types².

It is also difficult to understand why Ca²⁺ increases transiently within the cytosol and ER during addition of rotenone and oligomycin. During this phase of the protocol, Ca²⁺ is restored within depolarized mitochondria. The authors should give a plausible explanation for these subcellular Ca²⁺ fluxes.

We agree with the Reviewer that this is an interesting question. Calcium signaling plays a clear role in autophagy, but the details are still debated, as reviewed in [³]. The initial changes in calcium

levels are consistent with previous results in the literature, as discussed in the manuscript. The literature is less clear on the expected calcium dynamics and timescales of calcium signaling. The authors of the review paper³ point out that: “physiological cellular Ca²⁺ signals are typically brief, pulsatile events that do not trigger adverse outcomes.” They contrast this to many prior experiments probing the role of calcium signaling in autophagy, which used extended treatment with e.g. thapsigargin, which inhibits calcium pumping from the cytosol to the ER, or ionomycin, a calcium ionophore which allows calcium to leak through the plasma membrane. These extended changes in cellular calcium levels lead to clear changes in autophagy, but also perturb the cell in many other ways including ER stress, chronic depletion of intracellular Ca²⁺ stores, accumulation of unfolded proteins, mitochondrial depolarization, and fragmentation of organelles⁴⁻⁶.

We hypothesize that the observed transient calcium signals may have triggered the autophagy signaling cascade without the other effects of extended elevation in calcium levels. We now mention this possibility in the manuscript, though clearly a full mechanistic explanation would require substantial additional experiments.

Although it is beyond the scope of this paper, we believe the MOSAIC platform could help elucidate the complex dynamics of autophagy signaling. One could add additional sensors to record activity of key proteins in the signaling cascade, fluorescent assays of autophagy induction, and a more methodical exploration of starvation and pharmacology. One could use MOSAIC to track the relative timing of all these events.

In addition it might help to study and compare well-known Ca²⁺ signals in response to an activation of e.g. the IP3 pathway using ATP.

We agree that it would be interesting to explore calcium signaling (or MOSAIC signals more broadly) in HEK293 cells in response to ATP and many other perturbations. However, a detailed exploration of ATP responses in HEK293 cells is beyond the scope of this paper.

Ca²⁺ signals between the sub-compartments (cytosol, ER, Mitochondria) from the cell population might not correlate well due to cell-to-cell variations. Hence, some comparisons with single cell responses might be helpful.

We added Supplemental Video 1, which is paired with Figure 5, to examine heterogeneity in the cellular responses. We also added a paragraph of discussion at the end of the section *Probing metabolic perturbations*. This video enables a direct visualization of heterogeneity in cellular responses. As we mention in the new discussion, most sensors during most perturbations show little heterogeneity in their responses, although there are some exceptions. For cyto-, mito-, and ER-calcium there is good correlation between individual cells and the population average. If one is interested in cellular heterogeneity one can certainly quantify it from the MOSAIC data.

2.) Also in Figure 5: Why is the glucose level not reduced when glucose is removed from cells? Was glucose already removed prior to imaging?

Glucose was present prior to imaging. We also were surprised by the lack of response for the glucose sensor, given the robust response to a similar glucose deprivation in Supplemental Fig. S21. Given the low transfection efficiency for the glucose sensor in this particular array, we removed the glucose sensor time trace from Fig. 5.

3.) Measurement of mitochondrial ATP dynamics might represent a more informative response compared to cytosolic ATP changes; see Depaoli et. al Cell Reports 2018; DOI: 10.1016/j.celrep.2018.09.027.

We thank the reviewer for pointing this out and agree that mitochondrial ATP dynamics would be a valuable addition to future implementations of MOSAIC. We added a comment (and reference) for this and to two other relevant new fluorescent sensors in the Discussion.

Also for ATP changes huge cell-to-cell variations have been shown. The authors might better discuss that the low resolution (2 x objective) does not well allow to resolve single cell responses.

As mentioned above, we added a supplemental video and associated paragraph focused on heterogeneity in cellular responses. The microscope has 3 μm resolution, which is sufficient to resolve individual cellular responses (but not subcellular structures). As the video shows, the ATP responses were quite uniform in our experiments.

4.) The authors should cite Kuchenov et al., 2016, Cell Chemical Biology; <http://dx.doi.org/10.1016/j.chembiol.2016.11.008>. This work describes a similar approach using plasmids and FRET-based genetically encoded probes.

We thank the reviewer for bringing this citation to our attention. We added a paragraph to the discussion comparing the relative merits of printing plasmids and chemical transfection reagents vs lentivirus for preparing cellular microarrays. Chemical transfection, used in the above reference, is easier to implement and preferred for easy-to-transduce cell lines such as HEK293T cells. Viral transduction is necessary for hard-to-transfect cell lines such as cardiomyocytes.

Reviewer #2 (Remarks to the Author):

The authors have developed an impressive tool to parallelize a large number of optical measurements in a single system. The technique might be very useful for the field and is thoroughly characterized. Internal correction of the probe signals for pH effects is elegant, highly valuable and makes the setup robust.

We thank the Reviewer for the positive assessment.

While the technique itself is impressive and suitable for the journal, the biological relevance and results are not yet convincing and require some extra work. hPSC-CM experiments seem minimal. Furthermore, it should be made clear whether this tool is 'just' an all-in-one tool to measure multiple parameters or whether there is a benefit in measuring those parameters simultaneously – as is claimed.

Major points:

- Advantage 4 (“For cells that communicate over long distances via electrical signaling such as a cardiomyocytes or neurons, behavior can be synchronized throughout the well.”) does not take into account propagation times required for signals to travel. While this might not be problematic for slow response behaviour this will limit time-correlation between fast-response transients such as

AP, Ca and contractility. In turn, this might also provide an opportunity: is it possible to assess propagation, a key parameter for excitable cells, using this system?

Literature reports of conduction velocity in cultured CDI iCell cardiomyocyte syncytia vary, likely due to variations in cell density and culture maturity. Published propagation velocities are: 120 $\mu\text{m}/\text{ms}$ ⁷, 25 $\mu\text{m}/\text{ms}$ ⁸, 220 $\mu\text{m}/\text{ms}$ ⁹, 110 $\mu\text{m}/\text{ms}$ ¹⁰, and 70 $\mu\text{m}/\text{ms}$ ¹¹. If we take the median value of 110 $\mu\text{m}/\text{ms}$, it will take the voltage wave ~ 23 ms to propagate from islands on one edge of the array to the other edge of the array. At the 50 Hz frame-rate used in our MOSAIC recordings, this delay corresponds to ~ 1 frame. The edge-to-edge delay is $\sim 5\%$ of the shortest interpulse interval of ~ 500 ms (after β_2 -agonist addition, Fig. 7d & e), so the timing lag due to wave propagation will not impair matching of action potentials and calcium transients in different islands (see e.g. Figure 7d) We added a short discussion of the effects of conduction delay to the cardiomyocyte results section.

If conduction delays are a concern, one could readily put the voltage indicator islands on opposite edges of the MOSAIC array to quantify the conduction delay^{7,12}. We now mention this possibility in the section on iPSC cardiomyocytes.

- Details are required on the efficiency of transfection, especially when using hPSC-CMs since they are known to be not easily transfected. The manuscript should include transfection efficiency numbers so the number of cells from which the signals originate can be assessed. For example: cardiomyocytes are quite large so according to the calculations, islands might contain tens of cells. If the transfection efficiency is below 20% - which might not be an unrealistic estimate based on sensor 20 in figure 7b for example – this would result in maybe 30 cells * 20 percent = 6 cells. Since hPSC-CM populations are not entirely pure, supporting cell types such as fibroblasts might be the ones transfected instead of cardiomyocytes. More quantitative details on the transfection efficiency and identity of the target cells are required.

According to the literature, CDI iCell cardiomyocytes are $\sim 95\%$ pure¹³. In previous work in our lab, we grew iCell cardiomyocytes on fibronectin-coated polyacrylamide, as here⁷. Cells showed uniform staining for cardiac cytoskeleton marker cardiac troponin T across the extended syncytium, which revealed sarcomere banding in the contractile muscle fibers, suggesting a relatively pure cell population. Here is a representative image:

This should mitigate the reviewer's concern that the transduced cells are fibroblasts instead of cardiomyocytes.

Furthermore, we previously showed that lentivirus had ~90% transduction efficiency in hiPSC-derived cardiomyocytes^{7,14}.

There is further evidence in the data from this publication. The correlation images in Fig. 7c highlight pixels that co-vary in time with the average time trace of the two islands. That is, the correlation image highlights pixels that show a fluorescent signature of cardiomyocyte beating. Since only cardiomyocytes will show voltage and calcium transients synchronized with beating, the correlation image serves as a map of successfully transduced cardiomyocytes. We added language to the caption clarifying this. Time-correlated pixels show good transfection efficiency and expression across both islands.

We also added Supplemental Video 1, a time-lapse video of cardiomyocytes transduced by the printed cytosolic pHluorin sensor, which shows predominantly cardiomyocytes and a small fraction of migrating cells, likely fibroblasts. There are ~20 transduced cardiomyocytes per island. We added language describing the video, transduction efficiency, and number of transduced cells in the first paragraph of the section *MOSAIC in human iPSC-derived cardiomyocytes*.

The reviewer's estimate of tens of cardiomyocytes per island is accurate, and for the lower efficiency viruses such as the glucose sensor (pMOS020) that the reviewer mentioned, there can be 5 - 10 transduced cells per island. We added as sentence indicating this in the discussion of figure 6.

- Importantly, the results from the measurements with cardiomyocytes seem to be a single experiment and results are not quantitatively analysed. As standard in the field, this experiment should be repeated at least three times and claims such as changes in triangulation, oscillations, amplitudes, etc. should be made based on quantitative data from those repeated experiments. In addition, it should be corrected for baseline/vehicle measurements (wash in, wash out of buffers, etc.).

We now include examples of two other cardiac MOSAIC arrays (Fig. S23). As technical replicates they demonstrate that it is possible to create multiple high-quality cardiac MOSAIC arrays in which the cells beat reliably and the reporters are functional. Since these arrays were subjected to different pharmacological treatments, the physiological results are not compared across arrays.

We also added data (Fig. S24) to examine the effects of washing with buffer. We often observed transients during the initial two washes, but thereafter subsequent washes had little effect (see e.g. Fig. 7 and Figures S24 & S25). See the more detailed discussion in response to Reviewer 1 and the added sentence in the discussion of Fig. 5. Effects from washes were much smaller than effects from the adrenergic-modulating compounds shown in Fig. 7.

The primary purpose of the presented data is to demonstrate the potential of the MOSAIC platform for synchronized, multimodal recording. We carefully phrased the text so as not to make claims that were not supported by well-powered experiments.

- In line with the previous point, the authors mention "The increase in cytosolic calcium transient amplitude, expected for positive inotropic β 2-adrenergic stimulation⁷², did not occur immediately after perfusion but was observed in the subsequent high-speed recording 2 min later, implying that a signaling cascade was required to activate the inotropic effect." but also such an increase in AP amplitude is seen which is unexpected. How can the reader be sure this increase in amplitude is not

a washing artefact? Again, statistical comparison to baseline changes from multiple experiments are needed before making such claims.

See previous comment for a discussion of washing artifacts. The striking changes in beat rate and waveform shape in response to β_2 -adrenergic stimulation were much larger than any observed washing artifacts. The increase in beat rate and cytosolic calcium amplitude in response to β_2 -adrenergic stimulation has been well reported elsewhere, as discussed in detail in ref. ¹⁵ in the manuscript. We do not expect this result to be controversial.

The reviewer is surprised by the apparent changes in the voltage amplitude of the action potential. The AP amplitude decreased immediately by $\sim 20\%$ in response to β_2 -adrenergic stimulation in concert with the increase in beat rate (Fig. 7d). Two minutes later, when the calcium amplitude increased, the AP amplitude also increased by $\sim 10\%$. Prior experiments¹⁶ and simulations¹⁷ showed that β_2 -adrenergic activation can modestly increase AP amplitude via activation of I_{Ks} . We added a few sentences of discussion on this topic and we include four new references in the section *MOSAIC in human iPSC-derived cardiomyocytes*.

- In the discussion, the authors need to be more critically reflective on how their system compares to others. For example, how does it compare to other integrated optical solutions (For example Clyde Bioscience System and <https://www.nature.com/articles/s41467-019-12354-8>)?

We added a paragraph to the discussion section comparing MOSAIC, which can record tens of sensors in parallel, and spectral multiplexing, which can record up to 3 sensors in parallel. We cited the work of four different groups doing spectral multiplexing in cardiomyocytes, including the Reviewer's reference.

What conclusions can be drawn that cannot be drawn if the data are not measured simultaneously? Or is it merely increased throughput?

Having the relative timing of cellular signals provides critical information that is not available with independent, non-time-locked signals. Temporal correlation between signal implies that measured parameters are coupled, critical information in understanding cellular processes. Furthermore, the time-ordering of events helps elucidate the direction of causality in cellular signaling cascades.

In the presented data, there are several examples where time-locked, multiplexed recording leads to additional insight. For example:

1. In the time-correlated events in HEK293 cells in Fig. 5, we see that changes in cytosolic calcium and ATP levels precede changes in mito-calcium, ER-calcium, and NADH, suggesting that ATP and cyto-calcium are upstream in the signaling cascade.
2. Looking at the amplitude of calcium transients associated with action potentials in Fig. 7, one can see several examples where the cytosolic calcium transient amplitude changes but the mitochondrial transient does not, indicating that the two calcium levels are independently regulated. ER-calcium transient amplitudes, in contrast, more closely follow cytosolic amplitudes, suggesting that calcium leaving the ER is flowing directly to the cytosol.

3. The unexpected, correlated oscillations in baseline mito- and ER-calcium levels (Fig. 7e) suggest a connection between mitochondrial and ER calcium levels that is not mediated through cytosolic calcium.

We significantly expanded the discussion to highlight these features of the MOSAIC technique.

What are the limitations of the system?

We added a paragraph to the discussion discussing the limitation to MOSAIC and the challenges in its implementation.

Also, the analysis seems to require a lot of manual tweaking because of cross contamination. How does this affect the throughput?

We wrote a small graphical user interface (GUI) that steps island by island through the array and displays traces from the 16 subregions in each island. The human user can quickly override automated keep/discard calls for each subregion made by the default threshold. Manual curation of an island takes typically 5 seconds as the default calls are usually correct, although the conservative threshold often discards uncontaminated regions. In total, it takes ~5 minutes to curate a 42-island array, which is only a small fraction of the ~1 hour record times shown in the paper. As we make more MOSAIC recordings and build more confidence in thresholds and outlier detection, the manual curation step can likely be eliminated. For this first implementation, we erred on the side of caution. We added some of this information and discussion to the *Calculation of fluorescence time traces* subsection of the methods.

- Figure 7b, in 11 there seems to be a labelled region/area (or contamination??) stretching from one island to another one. What is happening here?

A contaminating hair or fiber adhered to the cells. This is rare but happens occasionally. We now comment on this in the text.

Minor points:

- Figure 2e and 2f are described in the text as 3e and 3f.

Corrected.

- In figure 7d, please add the names of the compounds. It is now a bit hard to follow.

Key added.

Reviewer #3 (Remarks to the Author):

To clarify simultaneously the different response behaviors in cells treated by chemical or physical treatment, the authors of this manuscript developed a technique, Multiplexed Optical Sensors in Arrayed Islands of Cells (MOSAIC). This method enables parallel sensing for different signaling or metabolism species or events utilizing the patterned cells encoded with different fluorescent sensor with microarray printing the sensor encoding lentiviral vectors. The special design for motile and nonmotile cells has also been displayed. This technique is a new advance in synchronous sensing of cell events in a number of living cells, which provides more accurate same culture conditions and

timing sequence. It is desirable the rapid publication in this journal. Before the acceptance for publication, the following concerns should be addressed.

We thank the Reviewer for the positive assessment.

1. The marker detection for the proposed autophagy in HEK293 cells upon nutrient deprivation should be carried out to confirm the hypothesis. Otherwise, the related discussion is questionable.

We tested upregulation in autophagy with an autophagic flux assay that uses a western blot to measure LC3-II levels, an autophagosome marker (Fig. S23). There was clear (>2-fold) and statistically significant upregulation in LC3-II in the full energy deprivation condition. In addition to the new supplemental figure, we mentioned this new data in the results section and updated the discussion.

2. What is exact reason for the different plasmid number in different islands, especially the islands for same sensing target? And how to guarantee the desired number in practical experiments.

By “plasmid number” we meant “plasmid identification number”, not “number of plasmids”. We have removed this ambiguity throughout the manuscript.

3. The immobilization of motile cells might be discussed further, how to evaluate the effect of immobilization and is there any other measures to improve the immobilization of the cells?

For HEK293T cells, we added some additional discussion under Fig. 3. As we believe the figure convincingly demonstrates, the HEK293T cells are very well confined to the patterned islands.

For cardiomyocytes, we originally patterned them into islands spanned by thin bridges, which we reasoned would make an electrically connected syncytium while limiting cell migration. However, after recording a 48-hour time-lapse video of the cardiomyocytes, we concluded that these cells do not migrate an appreciable amount. To simplify the experiment and create fully electrically connected sheet of cells, we switched to coating the entire polyacrylamide surface with fibronectin and patterning the lentivirus only (Fig. 1c). We added the time-lapse video to the supplemental information and describe it in the first paragraph under section *MOSAIC in human iPSC-derived cardiomyocytes*.

References

1. Rizzuto, R., Brini, M., Murgia, M. & Pozzan, T. Microdomains with high Ca²⁺ close to IP₃-sensitive channels that are sensed by neighboring mitochondria. *Science* (80-.). **262**, 744–747 (1993).
2. Ashrafi, G., de Juan-Sanz, J., Farrell, R. J. & Ryan, T. A. Molecular Tuning of the Axonal Mitochondrial Ca²⁺ Uniporter Ensures Metabolic Flexibility of Neurotransmission. *Neuron* **105**, 678-687.e5 (2020).
3. Bootman, M. D., Chehab, T., Bultynck, G., Parys, J. B. & Rietdorf, K. The regulation of autophagy by calcium signals: Do we have a consensus? *Cell Calcium* **70**, 32–46 (2018).
4. Sammels, E., Parys, J. B., Missiaen, L., De Smedt, H. & Bultynck, G. Intracellular Ca²⁺ storage in health and disease: A dynamic equilibrium. *Cell Calcium* **47**, 297–314 (2010).
5. Sakaki, K., Wu, J. & Kaufman, R. J. Protein kinase C θ is required for autophagy in response to stress in the endoplasmic reticulum. *J. Biol. Chem.* **283**, 15370–15380 (2008).
6. Ribeiro, C. M. P., McKay, R. R., Hosoki, E., Bird, G. S. J. & Putney, J. W. Effects of elevated

- cytoplasmic calcium and protein kinase C on endoplasmic reticulum structure and function in HEK293 cells. *Cell Calcium* **27**, 175–185 (2000).
7. Werley, C. A. *et al.* Geometry-dependent functional changes in iPSC-derived cardiomyocytes probed by functional imaging and RNA sequencing. *PLoS One* **12**, 1–22 (2017).
 8. Nguyen, C. *et al.* Simultaneous voltage and calcium imaging and optogenetic stimulation with high sensitivity and a wide field of view. *Biomed. Opt. Express* **10**, 789 (2019).
 9. Lee, P. *et al.* Simultaneous voltage and calcium mapping of genetically purified human induced pluripotent stem cell-derived cardiac myocyte monolayers. *Circ. Res.* **110**, 1556–1563 (2012).
 10. Li, J. *et al.* Extracellular recordings of patterned human pluripotent stem cell-derived cardiomyocytes on aligned fibers. *Stem Cells Int.* **2016**, 1–9 (2016).
 11. Dempsey, G. T. & Werley, C. A. Optogenetic Approach to Cardiotoxicity Screening: Simultaneous Voltage and Calcium Imaging Under Paced Conditions. in *Stem Cell-Derived Models in Toxicology* (eds. Clements, M. & Roquemore, L.) 109–134 (Springer Science+Business Media, 2017). doi:10.1007/978-1-4939-6661-5
 12. Werley, C. A., Chien, M.-P. & Cohen, A. E. An ultrawidefield microscope for high-speed fluorescence imaging and targeted optogenetic stimulation. *Biomed. Opt. Express* **8**, 5794 (2017).
 13. Kattman, S. J., Koonce, C. H., Swanson, B. J. & Anson, B. D. Stem cells and their derivatives: A renaissance in cardiovascular translational research. *J. Cardiovasc. Transl. Res.* **4**, 66–72 (2011).
 14. Hou, J. H., Kralj, J. M., Douglass, A. D., Engert, F. & Cohen, A. E. Simultaneous mapping of membrane voltage and calcium in zebrafish heart in vivo reveals chamber-specific developmental transitions in ionic currents. *Front. Physiol.* **5**, 1–10 (2014).
 15. Brodde, O.-E. The functional importance of beta1 and beta2 adrenoceptors in the human heart. *Am. J. Cardiol.* **62**, 24C-29C (1988).
 16. Volders, P. G. A. *et al.* Probing the contribution of IKs to canine ventricular repolarization: Key role for β -adrenergic receptor stimulation. *Circulation* **107**, 2753–2760 (2003).
 17. Heijman, J., Volders, P. G. A., Westra, R. L. & Rudy, Y. Local control of β -adrenergic stimulation: Effects on ventricular myocyte electrophysiology and Ca^{2+} -transient. *J Mol Cell Cardiol* **50**, 863–871 (2011).

Reviewers' Comments:

Reviewer #1:

Remarks to the Author:

Dear Authors,

Congratulations to this beautiful work. All my comments have been addressed satisfactorily.

Sincerely,

Roland Malli

Reviewer #2:

Remarks to the Author:

The authors have addressed most points adequately. It is regrettable they were unable to include biological repeats of their experiments on hiPSC cardiomyocytes and indeed, that only one (commercial) source of cardiomyocytes was used. This is correctly the current standard in the field since there is biological variability between lines (the authors actually cite the variability in conduction velocities they found in the literature for CDI cardiomyocytes) and batches although it is appreciated this is more of a "technical approach" paper and not intended to advance the biology. It is to be hoped then that the absolute values here are not cited verbatim by others.

Reviewer #3:

Remarks to the Author:

To clarify simultaneously the different response behaviors in cells treated by chemical or physical treatment, the authors of this manuscript developed a technique, Multiplexed Optical Sensors in Arrayed Islands of Cells (MOSAIC). This method enables parallel sensing for different signaling or metabolism species or events utilizing the patterned cells encoded with different fluorescent sensor with microarray printing the sensor encoding lentiviral vectors. The special design for motile and nonmotile cells has also been displayed. This technique is a new advance in synchronous sensing of cell events in a number of living cells, which provides more accurate same culture conditions and timing sequence. It is desirable the rapid publication in this journal.

Since all my concerns have been fully addressed, I suggest to publish it without no more revision.

Response to reviewer comments

Below we copy the Reviewer comments and give our replies in blue.

Dear Authors,

Congratulations to this beautiful work. All my comments have been addressed satisfactorily.

Sincerely,

Roland Malli

We thank the Reviewer for this positive assessment.

Reviewer #2 (Remarks to the Author):

The authors have addressed most points adequately. It is regrettable they were unable to include biological repeats of their experiments on hiPSC cardiomyocytes and indeed, that only one (commercial) source of cardiomyocytes was used. This is correctly the current standard in the field since there is biological variability between lines (the authors actually cite the variability in conduction velocities they found in the literature for CDI cardiomyocytes) and batches although it is appreciated this is more of a "technical approach" paper and not intended to advance the biology. It is to be hoped then that the absolute values here are not cited verbatim by others.

We amended the Discussion to note that quantitative analysis of these effects will require additional replicates of these experiments and comparison of cardiomyocyte cell lines from different sources.

Reviewer #3 (Remarks to the Author):

To clarify simultaneously the different response behaviors in cells treated by chemical or physical treatment, the authors of this manuscript developed a technique, Multiplexed Optical Sensors in Arrayed Islands of Cells (MOSAIC). This method enables parallel sensing for different signaling or metabolism species or events utilizing the patterned cells encoded with different fluorescent sensor with microarray printing the sensor encoding lentiviral vectors. The special design for motile and nonmotile cells has also been displayed. This technique is a new advance in synchronous sensing of cell events in a number of living cells, which provides more accurate same culture conditions and timing sequence. It is desirable the rapid publication in this journal.

Since all my concerns have been fully addressed, I suggest to publish it without no more revision.

We thank the Reviewer for this positive assessment.